# Pre-complexation of talin and vinculin without tension is required for efficient nascent adhesion maturation

Sangyoon J Han[1,2]*, Evgenia V Azarova[3], Austin J Whitewood[4], Alexia Bachir[5], Edgar Guttierrez[6], Alex Groisman[6], Alan R Horwitz[5†], Benjamin T Goult[4], Kevin M Dean[3]*, Gaudenz Danuser[1,3]*

[1]Lyda Hill Department of Bioinformatics, University of Texas Southwestern Medical Center, Dallas, United States; [2]Department of Biomedical Engineering, Michigan Technological University, Houghton, United States; [3]Department of Cell Biology, University of Texas Southwestern Medical Center, Dallas, United States; [4]School of Biosciences, University of Kent, Canterbury, United Kingdom; [5]Department of Cell Biology, University of Virginia, Charlottesville, United States; [6]Department of Physics, University of California San Diego, San Diego, United States

*For correspondence:
sjhan@mtu.edu (SJH);
Kevin.Dean@UTsouthwestern.edu (KMD);
gaudenz.Danuser@ utsouthwestern.edu (GD)

Present address: †Allen Institute for Cell Science, Seattle, United States

Competing interests: The authors declare that no competing interests exist.

**Abstract** Talin and vinculin are mechanosensitive proteins that are recruited early to integrin-based nascent adhesions (NAs). In two epithelial cell systems with well-delineated NA formation, we find these molecules concurrently recruited to the subclass of NAs maturing to focal adhesions. After the initial recruitment under minimal load, vinculin accumulates in maturing NAs at a ∼ fivefold higher rate than in non-maturing NAs, and is accompanied by a faster traction force increase. We identify the R8 domain in talin, which exposes a vinculin-binding-site (VBS) in the absence of load, as required for NA maturation. Disruption of R8 domain function reduces load-free vinculin binding to talin, and reduces the rate of additional vinculin recruitment. Taken together, these data show that the concurrent recruitment of talin and vinculin prior to mechanical engagement with integrins is essential for the traction-mediated unfolding of talin, exposure of additional VBSs, further recruitment of vinculin, and ultimately, NA maturation.

## Introduction

Cell-matrix adhesions are macromolecular complexes that link the extracellular matrix (ECM), typically via integrin transmembrane receptors, to the actin cytoskeleton. Being both a force-transmitter and a force-sensor, cell-matrix adhesions are critical to cell morphogenesis and mechanosensation (*Parsons et al., 2010*; *Discher et al., 2005*). Indeed, in response to ECM alterations, adhesions undergo constant changes in morphology and motion that involve the recruitment and recycling of a large number of adhesion molecules. For example, nascent adhesions (NAs) emerge within the actin-dense lamellipodia and then slide in the direction opposite of the protrusion as a result of polymerization-driven flow of the actin network (*Parsons et al., 2010*). Many of these NAs, which are less than 0.5 μm long (and thus appear as diffraction or near-diffraction limited spots in fluorescence microscopy) are rapidly turned over and disassembled. A subset of NAs matures into longer focal complexes (FCs, >0.5 μm in length) and focal adhesions (FAs,>2 μm in length) at the lamellipodia-lamella interface (*Parsons et al., 2010*; *Gardel et al., 2010*). During this progression, NAs go through multiple decision processes regarding fate and morphology. Compared to the well-studied FAs, for which the interconnection between structure, signaling, and force transmission is largely understood (*Kanchanawong et al., 2010*; *Plotnikov et al., 2012*; *Thievessen et al., 2013*; *Geiger et al., 2009*; *Chrzanowska-Wodnicka and Burridge, 1996*; *Han et al., 2012*; *Riveline et al.,*

*2001*; *Balaban et al., 2001*; *Stricker et al., 2011*), much less is known about the molecular and mechanical factors that determine NA assembly, turnover, and maturation. In part, this is because until recently, it has not been technically feasible to measure whether individual NAs transduce traction forces. By combining high refractive-index and mechanically tuned substrates that are compatible with total internal reflection microscopy (*Gutierrez et al., 2011*) and numerical methods for the computational reconstruction of cell-substrate traction at the single micron length-scale, we demonstrated that force transmission is essential for the stabilization and maturation of NAs (*Han et al., 2015*). However, it remains unknown which molecular factors determine whether a NA begins to bear forces and thus continues to assemble.

One possible factor that may influence NA maturation is the stoichiometry of the earliest molecular components recruited to the site of its formation (*Zaidel-Bar et al., 2004*; *Digman et al., 2009*). In particular, the recruitment of talin, vinculin, and paxillin could play a critical role as they all are known to be mechanosensitive (*Austen et al., 2015*; *Humphrey et al., 2014*; *del Rio et al., 2009*; *Kumar et al., 2016*; *Carisey et al., 2013*; *Humphries et al., 2007*; *Schiller et al., 2011*; *Pasapera et al., 2010*). Talin is an integrin activator (*Moser et al., 2009*; *Tadokoro et al., 2003*) that directly links integrins to the actin cytoskeleton (*Calderwood et al., 2013*). Under force, the helix bundle domains in talin's rod-like region unfold (*del Rio et al., 2009*), which both disrupts ligand binding and exposes cryptic-binding sites for vinculin and other proteins (*del Rio et al., 2009*; *Yao et al., 2016*; *Yan et al., 2015*; *Goult et al., 2013*; *Goult et al., 2018*). Vinculin, when bound to talin's exposed binding sites, can indirectly strengthen the connection between actin and integrins by (1) forming a catch bond with F-actin (*Case et al., 2015*; *Huang et al., 2017*), (2) establishing multivalent linkages between talin and F-actin (*Yao et al., 2016*; *Yan et al., 2015*; *Atherton et al., 2015*), and (3) stabilizing talin's unfolded state (*Yao et al., 2014*). In this scenario, talin first binds to integrins and F-actin, is unfolded under the initial load, and serves as a scaffold for the recruitment of vinculin. Indeed, at the level of FAs, direct evidence for catch-bonds (*Bell, 1978*; *Thomas, 2008*; *Thomas et al., 2008*), and the exposure of hidden binding sites under load (*Vogel and Sheetz, 2006*; *Zhu et al., 2008*), has established the idea of force-assisted adhesion growth. Further evidence for this model indicates that downregulation of actomyosin contractility reduces the recruitment of vinculin (*Pasapera et al., 2010*) and other adhesion proteins (*Kuo et al., 2011*), as well as the association between talin and integrins (*Bachir et al., 2014*).

In contrast to a model of hierarchical FA growth and stabilization where talin arrives first and recruits vinculin, fluorescence fluctuation analyses (*Bachir et al., 2014*) and co-immunoprecipitation experiments (*Pasapera et al., 2010*) have suggested that talin and vinculin might form a complex before talin associates with integrins. While talin-vinculin pre-association implies vinculin's force-independent binding to talin, it is not clear whether this pre-association is required for NA assembly, and if so, whether pre-association affects the decision processes for NA maturation. Moreover, paxillin, a scaffolding protein that works in close relationship with focal adhesion kinase (FAK) (*Pasapera et al., 2010*; *Parsons, 2003*; *Schlaepfer and Mitra, 2004*; *Mitra and Schlaepfer, 2006*), is thought to be recruited and stabilized by force at an early phase of NA assembly (*Plotnikov et al., 2012*; *Schiller et al., 2011*; *Choi et al., 2008*; *Deakin and Turner, 2008*). However, vinculin's recruitment and role in establishing tension across NAs remains poorly understood (*Laukaitis et al., 2001*; *Webb et al., 2004*; *Wiseman et al., 2004*).

Here, we investigated the integration of molecular recruitment and mechanical forces in determining the fate of NAs. Specifically, we combined high-resolution traction force microscopy with highly sensitive particle detection and tracking software to evaluate the time courses of force transmission and the molecular composition of individual NAs. A comprehensive inventory of these traces revealed broad heterogeneity in NA behavior. Thus, we applied machine-learning approaches to divide NAs into subgroups with distinct characteristics and identified that five subgroups are necessary to account for the different kinematic, kinetic, and mechanical properties of NAs. By focusing on the NA subgroup that matures into stable FAs, we found that the pre-complexation of talin and vinculin under force-free conditions was mediated by talin's R8 domain. These complexes strengthen the link between talin and F-actin, aid the unfolding of talin upon initial weak force transmission in order to expose additional vinculin-binding sites, and ultimately support the transition of spontaneously formed nascent adhesions into stable focal adhesions.

## Results

### Nine adhesion classes can be distinguished based on different kinetic and kinematic behaviors

To investigate the time courses of traction forces and protein recruitment in NAs, we performed two-channel time-lapse, total internal reflection fluorescence (TIRF) imaging of Chinese Hamster Ovary epithelial cells (ChoK1, *Figure 1—figure supplement 1a*). We chose these cells because they form a stable footprint in close apposition to a flat substrate, which is ideal for TIRF microscopy of individual adhesions (*Digman et al., 2009*; *Bachir et al., 2014*; *Bate et al., 2012*; *Macdonald et al., 2008*). In addition, these cells display a relatively slow progression of adhesion assembly, offering the opportunity to track the fate of NAs and to identify the requirements for their maturation (*Choi et al., 2008*; *Laukaitis et al., 2001*; *Webb et al., 2004*; *Choi et al., 2011*). ChoK1 cells were transiently transfected with low-levels of either talin-GFP, vinculin-GFP, or paxillin-GFP, and plated and allowed to spread and migrate on high-refractive index silicone gels. To measure traction forces, deformable silicone gels were densely coated with 40 nm fluorescent beads ($1.54 \pm 0.22$ beads/$\mu$m, $0.42 \pm 0.17$ $\mu$m bead-to-bead spacing, *Figure 1—figure supplement 1b,c*). After each experiment, cells were removed from the substrate and the beads were imaged in the relaxed, undeformed configuration of the silicone gel, which permits the quantitative reconstruction of traction forces at submicron scales (*Figure 1—figure supplement 1e,g,h*; *Han et al., 2015*). As expected, all traction force vectors pointed from the cell periphery to the cell center, independent of which adhesion protein was imaged (*Figure 1a–c*). Regardless of which protein was ectopically expressed, ChoK1 cells spread and generated traction forces indistinguishably (*Figure 1—figure supplement 2*).

Fluorescently tagged adhesion proteins (*Figure 1d–f*) were detected and tracked, and their intensity time courses extracted from their trajectories (*Figure 1g*, *Figure 1—figure supplement 1f*). To account for the heterogeneity of adhesions, we collected 22 features from each trajectory (*Figure 1h*, *Supplementary file 1A*) and classified the adhesions into nine groups (see *Supplementary file 1B* for a summary of each group) with a supervised machine learning pipeline (see Methods, Software Availability). To generate training data, a human operator used a dedicated graphical user interface for labeling ~120 adhesion tracks (~10 tracks per group, out of ~10,000 tracks per movie). Based on these data we trained a support vector machine (SVM) classifier (validation accuracy: 70–80%, *Figure 1—figure supplement 3a*). All features were inspected for redundancy and similarity (*Figure 1—figure supplement 3b–c*), and each group was distinct in terms of its Euclidean distance to the closest group in feature space (*Figure 1—figure supplement 3d*). SVM-based classification of trajectories that were excluded from the training data assigned each adhesion to one of nine different classes, G1, G2, . . ., G9 (*Figure 1i*). Five of the nine classes (G1-G5) identified NAs, three (G6-G8) identified FAs, and one group (G9) contained insignificant, noise-like trajectories (*Video 1*). The five NA classes significantly differed in terms of features such as 'edge protrusion speed' (*Figure 1j*), 'adhesion movement speed' (*Figure 1k*), 'average fluorescence intensity' (*Figure 1l*), and 'lifetime' (*Figure 1m*). For example, NAs classified as G3 form at the tip of the protruding edge and move forward with the protrusion. Of all NA classes, their fluorescence intensity is the lowest (*Figure 1l*). NAs classified as G2 form at the protruding edge but slide rearward relative to the substrate, mature to form larger FCs or FAs, and have the highest intensity and longest lifetime (*Figure 1l–m*). Indeed, all G2 NAs (~100%) mature to FCs, and ~32% mature into FAs (*Figure 1—figure supplement 4*). NAs classified as G1 also form at the protruding edge, but they are relatively stationary (*Figure 1k*) and are characterized by a weak fluorescence intensity and a short lifetime (*Figure 1l–m*).

### Nine adhesion classes exhibit distinct mechanical behaviors

Next, we tested the hypothesis that these spatially and kinetically distinct classes of NAs generated differential traction forces. Indeed, we found that the subgroup of maturing NAs, G2, showed the highest traction magnitude shortly after their initial assembly (*Figure 1n*). This is consistent with previous findings that demonstrated the tension-mediated and myosin-dependent maturation of FAs (*Schiller et al., 2011*; *Choi et al., 2008*; *Schiller et al., 2013*). Interestingly, NAs in G3 exhibited an insignificant amount of traction that did not increase throughout their lifetime (*Figure 1n*, *Figure 1—figure supplement 5*), suggesting that this population of anterogradely moving NAs might not

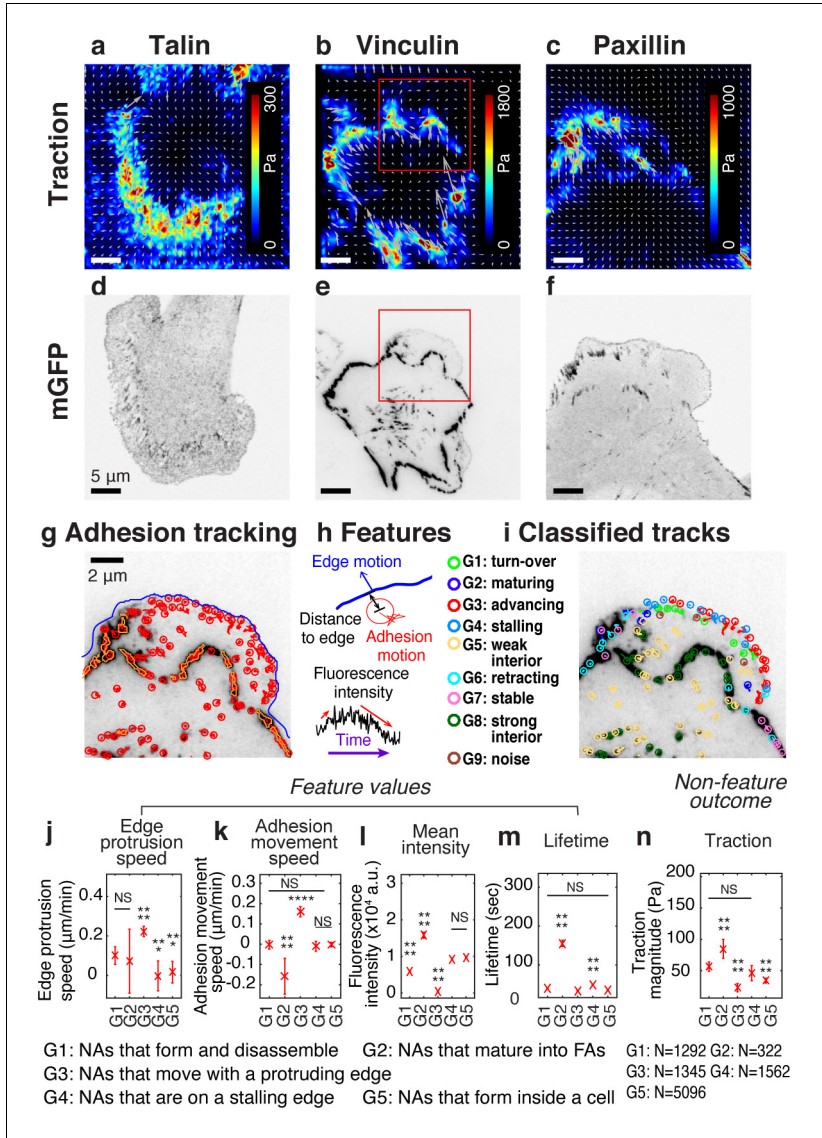

**Figure 1.** Experimental/computational pipeline to analyze heterogeneous adhesion dynamics in ChoK1 cells. (**a–c**) High-resolution traction maps co-imaged with mGFP-tagged adhesion protein, talin (**d**), vinculin (**e**), and paxillin (**f**). 5 kPa silicone gel coated with high density beads was used as a TFM substrate. (**g**) Trajectories of individual nascent and focal adhesions overlaid on a region of interest cropped from (**e**). Tracking is based on all detected point sources, (red circles). Big segmented focal contacts/adhesions (orange, closed freeform overlays) were used as additional information for feature selection. (**h**) Some of the key features used for supervised classification, tabulated in *Supplementary file 1A*. (**i**) Classification of adhesion trajectories into nine different groups, overlaid on the adhesion image. Five different NA groups, three FA groups and one noise group were distinguished by the support vector machine classifier. (**j–m**) Comparison of feature values among the five NA groups, G1, G2, G3, G4, and G5: edge protrusion speed (**j**), adhesion movement speed, positive when sliding toward protruding edge (**k**), mean intensity (**l**), and lifetime (**m**), extracted from six vinculin-tagged cells. All features show a significant shift in value for at least one subgroup. (**n**) Average traction magnitude, read from traction map, at individual NA trajectories per each group. The number of samples per each group is shown in the lower right corner of the figure.

The online version of this article includes the following source data and figure supplement(s) for figure 1:

**Source data 1.** Source data for *Figure 1j*.
**Source data 2.** Source data for *Figure 1k*.
**Source data 3.** Source data for *Figure 1l*.
**Source data 4.** Source data for *Figure 1m*.
**Source data 5.** Source data for *Figure 1n*.

*Figure 1 continued on next page*

*Figure 1 continued*

**Figure supplement 1.** Simultaneous TFM-adhesion experimental approach.
**Figure supplement 2.** Overall average traction per cell and cell spreading area did not change with expression of talin-GFP, vinculin-GFP, or paxillin-GFP.
**Figure supplement 3.** Validation of SVM-based machine learning.
**Figure supplement 4.** Boxplot of the fraction of G2 NAs that mature into FCs, FAs, or either FCs and FAs.
**Figure supplement 5.** Representative time series of fluorescence intensity, traction magnitude, edge protrusion speed, adhesion sliding speed, and distance to closest edge, for the five NA groups (G1–G5).
**Figure supplement 6.** Differences in the feature values for NA subgroups in paxillin and talin time-lapse images.
**Figure supplement 7.** Classification shifts due to substrate stiffness.

interact with the retrogradely flowing F-actin cytoskeleton. NAs in G1 had higher traction than those in G3, implicating that short-lived, non-maturing NAs can transmit significant amounts of traction forces, which is consistent with our previous findings (*Han et al., 2015*). Importantly, these trends were consistent regardless of which adhesion protein was used for tracking (talin, vinculin, or paxillin; *Figure 1—figure supplement 6*). Furthermore, we observed shifts in the relative proportion of adhesion classes when cells were cultured on stiffer substrates (*Figure 1—figure supplement 7*). For example, large FAs (G8) were more abundant in ChoK1 cells cultured on 18 kPa substrates when compared to ChoK1 cells on 5 kPa substrates, which is a stereotypical output for stiffness sensing across adhesions (*Han et al., 2012*; *Elosegui-Artola et al., 2016*). Likewise, many populations of NAs (i.e. those in G1, G2, and G4) decreased for cells on 18 kPa relative to their 5 kPa counterparts. Altogether, these results confirm the reliability of the SVM classifier and suggest that kinetically unique NAs also show differences in terms of force transduction.

## Talin, vinculin, and paxillin are recruited sequentially in non-maturing NAs, but concurrently in maturing NAs with traction development

To evaluate the relationship between molecular recruitment and traction force in NAs, we performed high-resolution traction force microscopy on cells labeled with either talin, vinculin, or paxillin, and

processed these data using the aforementioned SVM classifier (*Videos 1–9*). We focused our analysis on the differences between NAs in G1 and G2 (*Figure 2*), which are for simplicity henceforth referred to as non-maturing and maturing NAs, respectively. For all proteins evaluated, fluorescence intensity traces for non-maturing NAs had a lifetime of ~6–7 min, with distinct phases of rise and fall (*Figure 2a–f*, top). The associated traction traces exhibited intermittent rises and falls, but with an overall magnitude that was much smaller than the traction traces in maturing NAs (*Figure 2a–f*, bottom). As expected, maturing NAs showed a steady increase in both fluorescence intensity and traction, and had a lifetime greater than 15 min (*Figure 2g–l*). The fluorescence intensity and traction of individual non-maturing and maturing NAs reflected this stereotypical behavior, and so did the average behavior, that is a slight increase and fall for non-maturing NAs, and more steady increase for maturing NAs (*Figure 2—figure supplement 1a–i*). A further analysis with cohort plots, where traces with similar lifetimes are grouped and displayed separately, revealed that average traces of many cohorts follow the

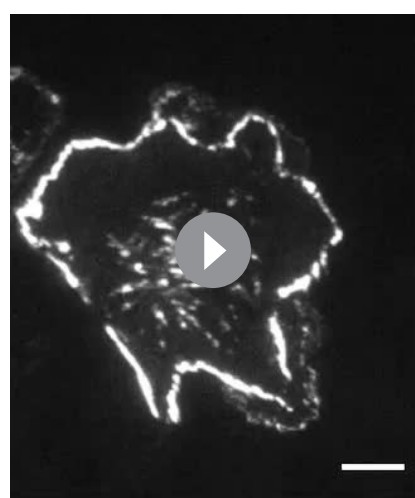

**Video 1.** Time-lapse images of GFP-tagged vinculin in a ChoK1 cell, overlaid with adhesion trajectories and classification state from the support vector machine (SVM)-based machine-learning. Different colors represent different classes: G1 (light green), G2 (dark blue), G3 (red), G4 (light blue), G5 (yellow), G6 (cyan), G7 (pink), G8 (dark green), and G9 (brown). The time interval per frame: 2 s. Playing speed: 25 frames/s. Duration of the movie: 12 min. Scale bar: 5 µm.
https://elifesciences.org/articles/66151#video1

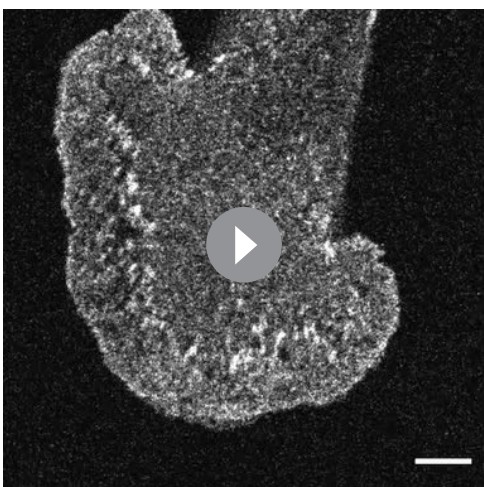

**Video 2.** Time-lapse images of GFP-tagged talin in a ChoK1 cell. The time interval per frame: 1.644 s. Playing speed: 25 frames/s. Duration of the movie: 8 min 13 s. Scale bar: 5 μm.
https://elifesciences.org/articles/66151#video2

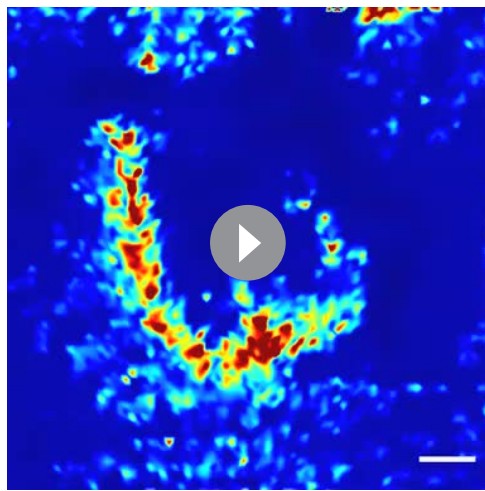

**Video 3.** Time-lapse images of traction force magnitude for a ChoK1 cell expressing GFP-talin. Traction forces are reconstructed from the bead images using high-resolution, L1-regularized, TFM software (*Han et al., 2015*). The color scale is the same as one at *Figure 1a*, that is, 0 (blue) – 300 Pa (red). The time interval per frame: 1.644 s. Playing speed: 25 frames/s. Duration of the movie: 8 min 13 s. Scale bar: 5 μm.
https://elifesciences.org/articles/66151#video3

stereotypical behavior (*Figure 2—figure supplement 1j–o*).

Next, we developed an event-based time-series analysis method that identifies the first time point of significant fluorescence and force increase, respectively, and then measures the time delay between the two (*Figure 2m*). The blue and red arrows in *Figure 2d–f,j–l* show, in two example traces, the time points identified statistically as the initial time point of intensity and traction increase, respectively. Using this approach, we first determined the fraction of NAs per group with a significant traction increase at any point throughout their lifetime (*Figure 2—figure supplement 2*). Interestingly, both non-maturing and maturing NAs showed such force increases, that is, they were engaging and clutching against F-actin's retrograde flow at one point with the substrate. In contrast, NAs classified as groups G3-5 exhibited lower fractions of force increase, suggesting that a very large number of adhesion protein complexes, detectable through either talin, vinculin, or paxillin recruitment, do not transduce measurable traction forces (e.g. less than 20 Pa). Indeed, these complexes could result from an incomplete molecular clutch and by being engaged with the substrate but not F-actin, or vice versa.

To evaluate if maturing and non-maturing NAs were differentially assembled, we next analyzed protein recruitment using the initial traction force increase as a time fiduciary. In non-maturing NAs, talin, vinculin and paxillin were recruited ~18 s, ~8 s and ~4 s before the onset of force transmission, respectively (*Figure 2n*). In contrast, in maturing NAs, talin, vinculin and

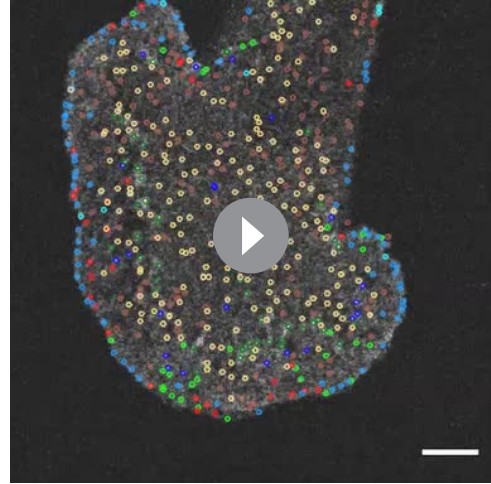

**Video 4.** Time-lapse images of GFP-tagged talin in a ChoK1 cell, overlaid with adhesion trajectories and classification state from the support vector machine (SVM)-based machine-learning. The color coding is the same as in the legend of *Video 1*. The time interval per frame: 1.644 s. Playing speed: 25 frames/s. Duration of the movie: 8 min 13 s. Scale bar: 5 μm.
https://elifesciences.org/articles/66151#video4

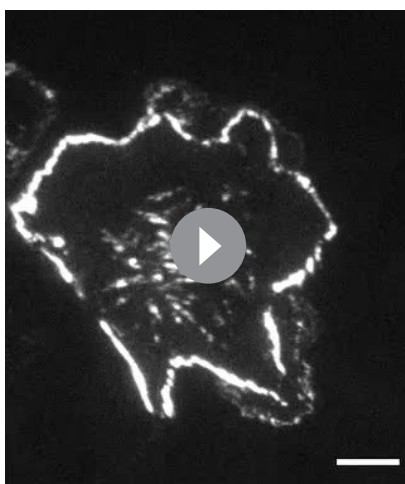

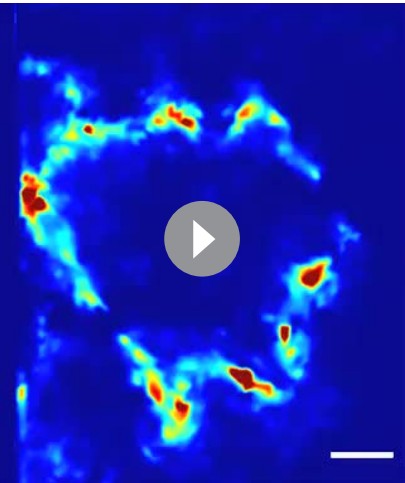

**Video 5.** Time-lapse images of GFP-tagged vinculin in a ChoK1 cell. The time interval per frame: 2 s. Playing speed: 25 frames/s. Duration of the movie: 12 min. Scale bar: 5 µm.

https://elifesciences.org/articles/66151#video5

**Video 6.** Time-lapse images of traction force magnitude for a ChoK1 cell expressing GFP-vinculin. Traction forces are reconstructed from the bead images using high-resolution, L1-regularized, TFM software (*Han et al., 2015*). The color scale is the same as one at *Figure 1b*, that is, 0 (blue) – 1800 Pa (red). The time interval per frame: 2 s. Playing speed: 25 frames/s. Duration of the movie: 12 min. Scale bar: 5 µm.

https://elifesciences.org/articles/66151#video6

paxillin were recruited concurrently ~4 s before the onset of force transmission (*Figure 2o*). We also noted that the temporal distribution of talin recruitment was significantly wider in non-maturing NAs than in maturing NAs. These findings suggest that in maturing NAs talin localizes with vinculin prior to its association with integrins, which more efficiently leads to force transmission and maturation to a FC. These data also suggest that NAs that sense force at the earliest stages of assembly are more likely to successfully mature. In contrast, although non-maturing NAs also support some lower level of force transmission eventually (*Figure 1n*), talin and vinculin assemble sequentially and remain under force-free conditions for a much longer duration of time.

### In maturing NAs, vinculin assembles faster than in non-maturing NAs, but talin and paxillin show no difference

The rod domain of talin contains 13 helical bundles, 9 of which include cryptic vinculin binding sites (VBSs) that are exposed upon tension-mediated unfolding (*Geiger et al., 2009*; *del Rio et al., 2009*; *Goult et al., 2013*). As such, we hypothesized that the simultaneous recruitment of talin and vinculin in maturing NAs could further accelerate vinculin binding compared to non-maturing NAs. To test this, we quantified the assembly rate of each protein by obtaining the slope of the fluorescence intensity over the first 10 s after initial appearance (*Figure 3a*). Interestingly, only vinculin showed a significant difference in the assembly rate between non-maturing vs. maturing vinculin complexes, while talin and paxillin showed no such differences (*Figure 3a*). Thus, the concurrent arrival time of talin and vinculin in maturing NAs could prime

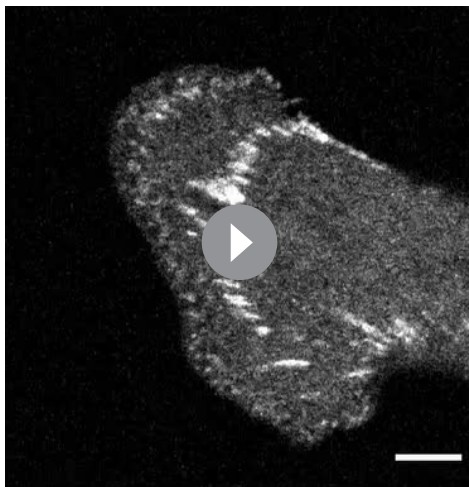

**Video 7.** Time-lapse images of GFP-tagged paxillin in a ChoK1 cell. The time interval per frame: 1.644 s. Playing speed: 25 frames/s. Duration of the movie: 8 min 13 s. Scale bar: 5 µm.

https://elifesciences.org/articles/66151#video7

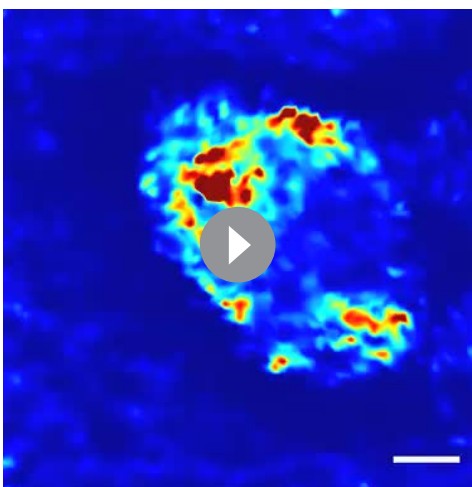

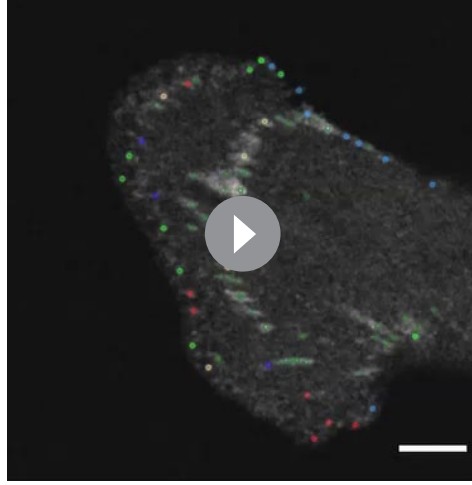

**Video 8.** Time-lapse images of traction force magnitudes generated by a ChoK1 cell expressing GFP-paxillin. Traction forces are reconstructed from the bead images using high-resolution, L1-regularized, TFM software (*Han et al., 2015*). The color scale is the same as one at *Figure 1c*, that is, 0 (blue) – 1000 Pa (red). The time interval per frame: 1.644 s. Playing speed: 25 frames/s. Duration of the movie: 8 min 13 s. Scale bar: 5 μm.

https://elifesciences.org/articles/66151#video8

**Video 9.** Time-lapse images of GFP-tagged paxillin in a ChoK1 cell, overlaid with adhesion trajectories and classification state from the support vector machine (SVM)-based machine-learning. The color coding is the same as in the legend of *Video 1*. The time interval per frame: 1.644 s. Playing speed: 25 frames/s. Duration of the movie: 8 min 13 s. Scale bar: 5 μm.

https://elifesciences.org/articles/66151#video9

talin to expose additional VBS domains, which in turn would further reinforce vinculin recruitment and adhesion maturation. We also quantified the traction force growth in those NAs with an expectation that there would be an immediate rise in force with faster vinculin binding. However, the traction force growth rate for the first 10 s showed no significant difference between non-maturing and maturing NAs (*Figure 3b*). This observation could arise because the forces were below our limits of detection for traction, or because traction does not develop during the earliest stage of molecular recruitment. N, a difference was observed when the force growth rate was quantified over a longer period, that is, 2 min (*Figure 3c*), which is consistent with our previous observations (*Han et al., 2015*). These findings imply that increased vinculin recruitment in maturing NAs supports the rise in traction force by increasing binding to F-actin, albeit with some time delay.

## Vinculin can bind to talin without force through a 'threonine belt' in talin R8 domain

Previous work established that under tension the R3 domain in talin unfolds first, as it contains a destabilized hydrophobic core due to the presence of a 'threonine belt'. By mutating the threonine residues to isoleucines and valines (the so called 'IVVI mutant') it was possible to stabilize the core and prevent talin from unfolding, which significantly reduces the exposure of the two cryptic VBS (*Goult et al., 2013*; *Yao et al., 2014*; *Elosegui-Artola et al., 2016*). Moreover, we showed that the VBS in R8 was able to bind vinculin readily in the absence of force (*Gingras et al., 2010*). Like R3, R8 also contains a threonine belt, consisting of T1502, T1542, and T1562 (*Figure 4a*). Thus, we hypothesized that a similar mutational strategy, using a T1502V, T1542V, T1562V 'R8vvv mutant', could stabilize the R8 domain and reduce access to the VBS. To test this hypothesis, we made a 'R7R8vvv' construct and compared its unfolding characteristics to the wild-type (WT) R7R8 fragment (R7R8wt) with circular dichroism (CD; *Figure 4b*). We included the R7 domain to improve the stability of the fragment and maintain R8 in its native conformation. In the R7R8wt the two domains unfolded cooperatively with a single unfolding step at a melting temperature ($T_m$) of 55°C. In contrast, stabilization of the R8 domain in the R7R8vvv mutant resulted in the domains unfolding independently, with R7 unfolding at a similar temperature to the WT ($T_m$56°C), but the melting temperature of the R8 domain increased from 56°C to 82°C. Strikingly, the two unfolding steps indicate that in the R7R8vvv

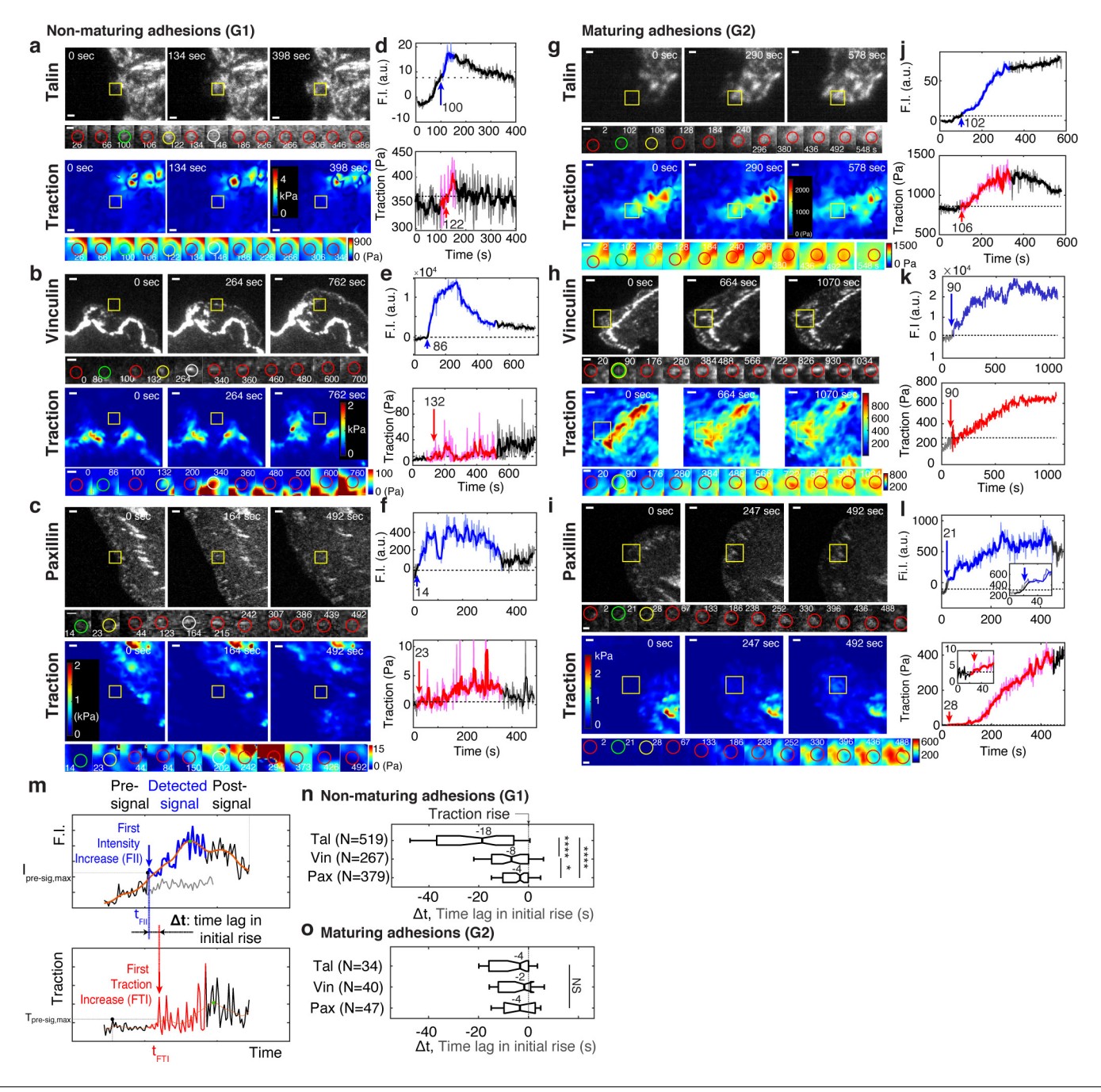

**Figure 2.** Talin and vinculin in non-maturing NAs are recruited in a sequential manner before traction development whereas in maturing NAs they are recruited concurrently, along with paxillin, briefly before the initial traction rise. (a–l) Representative traces of protein recruitment and traction generation at non-maturing (a,b,c), or maturing (g,h,i) NAs. Each panel contains three views at different time points of mGFP-tagged talin (a,g, top), vinculin (b,h, top), and paxillin (c,i, top) and associated traction maps (bottom). Yellow boxes indicate positions of example adhesions whose fluorescent signals and traction levels are shown in time lapse montages with finer resolution underneath. Scale bar: 1 μm. Green circle represents the time point of initial talin/vinculin/paxillin signal rise, yellow circles show the time point of initial traction rise, and white circles shows the time of the peak amplitude, while red circles show regular detections in between these events. Scale bar: 1 μm. (d–l) Traces of fluorescence intensity amplitude (top) and traction (bottom). Blue and red segments indicate periods of significant fluorescence intensity amplitude and of traction, respectively, illustrated as 'detected signal' in (m). The black segments indicate the remaining background-subtracted fluorescence intensity and traction levels outside the detected signal period, that is, pre-signal and post-signal illustrated in (m). Whereas colored segments are read at positions moving with the adhesion center, pre- and post-signal traces are read at the first and last position of detected signal. An inset in (l) indicates that also in this trace the traction is gradually increasing. Blue and red arrows with the numbers on top mark the time points in seconds of the first intensity increase (FII) and the first traction

*Figure 2 continued on next page*

*Figure 2 continued*

increase (FTI), respectively, which are defined in (m). (m–o) Analysis of time-shifts between protein recruitment and FTI. (m) Traces of fluorescence intensity (top) and traction (bottom). Illustrated is the detection of the first significant value in both series. The gray signal represents the local background around the detected NA. Distinct distributions of time lags between FII and FTI in non-maturing (n) and maturing (o) NAs. Sample numbers, extracted from six cells for talin, five cells for vinculin and four cells for paxillin, are shown with each y-axis label. *p<1×10$^{-2}$, ****p<1×10$^{-30}$ by Mann-Whitney U test.

The online version of this article includes the following source data and figure supplement(s) for figure 2:

**Source data 1.** Source data for *Figure 2n*.
**Source data 2.** Source data for *Figure 2o*.
**Figure supplement 1.** The time-series of fluorescence intensity and traction force within non-maturing (G1) and maturing (G2) NAs for talin, vinculin and paxillin show heterogeneity but 'increasing' trend within their lifetimes.
**Figure supplement 2.** Box plots of fractions of force-transmitting NAs for each classification group for (a) talin, (b) vinculin, and (c) paxillin.

mutant the R7 and R8 behave independently with regard to thermal stability. Together, these results show that the R7R8vvv mutant stabilizes R8.

To test whether stabilization of R8 would affect its interaction with vinculin, we used analytical gel filtration to look at protein complex formation. Preincubation of R7R8wt or R7R8vvv with vinculin Vd1 showed that both constructs were able to form complexes. However, the R7R8vvv:Vd1 complex peak was substantially smaller than the WT:Vd1 peak, confirming that the R7R8vvv bound less Vd1 (*Figure 4c*). Thus, by stabilizing the threonine belt with the R8 valine mutations, vinculin's accessibility to the cryptic VBS was reduced. Interestingly, the R8 domain is also a Leucine-Aspartic acid (LD) motif binding site, that is it binds to multiple proteins that act downstream of adhesion, including RIAM and DLC1 (*Goult et al., 2013*; *Goult et al., 2018*; *Zacharchenko et al., 2016*). Using a fluorescence polarization assay described previously (*Whitewood et al., 2018*), we measured the binding affinities of R8 ligands RIAM TBS1 and DLC1 peptides with R7R8wt and R7R8vvv (*Figure 4—figure supplement 1*). Both peptides showed comparable binding affinities toward WT R7R8 and the R8R7vvv mutant, confirming that R7R8vvv was still able to bind LD-motif proteins. Altogether, these biochemical characterizations suggest that the threonine belt in talin R8 exposes the R8 cryptic VBS and binds vinculin under force-free conditions. The mutant also provided a tool for us to probe

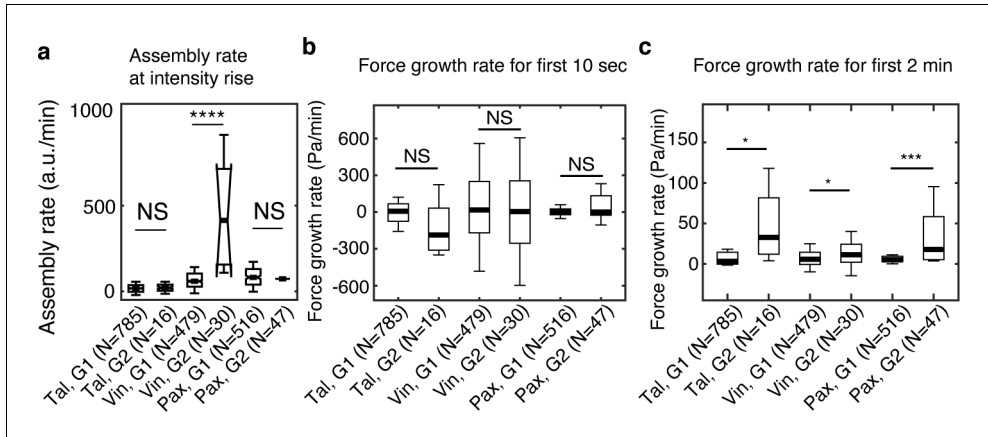

**Figure 3.** Vinculin, but not talin and paxillin, is recruited significantly faster in maturing NAs than in non-maturing NAs. (a) Assembly rate of talin, vinculin, and paxillin, to G1 (non-maturing) or to G2 (maturing) NAs, quantified by the slope of fluorescence intensity over the initial 10 s after detection. (b) Traction growth rate at the NAs in (a) for the initial 10 s after detection. (c) Traction growth rate quantified over the first 2 min after detection. *p<1×10$^{-2}$, ***p<1×10$^{-10}$, ****p<1×10$^{-30}$ by Mann-Whitney U test.

The online version of this article includes the following source data for figure 3:

**Source data 1.** Source data for *Figure 3a*.
**Source data 2.** Source data for *Figure 3b*.
**Source data 3.** Source data for *Figure 3c*.

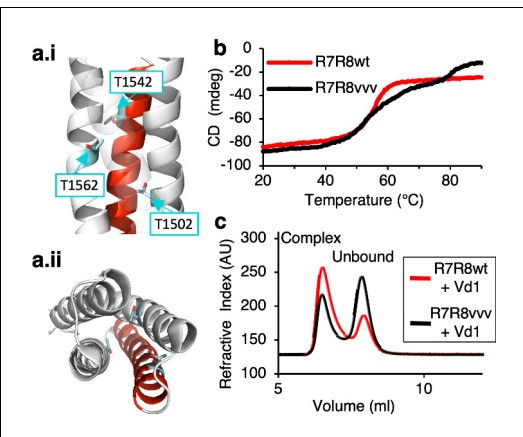

**Figure 4.** Stabilizing the 'threonine belt' in the R8 domain of talin inhibits talin-vinculin interactions under tension-free conditions. (**a**) Cartoon representation of talin R7R8 (pdb id 2X0C) showing the 'threonine belt', comprised of residues T1502, T1542, and T1562, labeled and shown as sticks (cyan), the VBS helix is colored red. (**a.i**) side on view (N.B. helix 31 transparent), (**a.ii**) top down view. (**b**) Denaturation profiles for WT R7R8wt (red) and R7R8vvv (black) measured by monitoring the change in circular dichroism at 208 nm with increasing temperature. R7R8wt has a melting temperature of 55°C, whereas R7R8vvv unfolds in two steps, one (R7) with a melting temperature of 56°C and R8 unfolding at 82°C. (**c**) Chromatograms showing binding of talin R7R8 to the vinculin head (Vd1). R7R8wt (red) and R7R8vvv (black) binding to Vd1. Complex peaks and unbound peaks are indicated.

The online version of this article includes the following figure supplement(s) for figure 4:

**Figure supplement 1.** Fluorescence polarization assay showing the binding affinities for R8 ligand peptides from (top) RIAM TBS1 and (bottom) DLC1 with WT and R7R8vvv.

the functional implications of talin-vinculin force-free complex formation on NA assembly and maturation in vivo without interfering with binding of other binding partners.

## Cells with R8vvv mutant talin show less maturing NAs and sparser and smaller FAs

To investigate whether talin's ability to form a complex with vinculin under force free conditions promotes adhesion maturation, we introduced the R8vvv mutation into full-length talin1 and tagged it with mNeonGreen ('talin1 R8vvv-mNG'). For imaging, we knocked down endogenous talin1 expression with an shRNA, and rescued ChoK1 cells with shRNA-resistant forms of WT or R8vvv talin. The expression of talin1 R8vvv-mNG was slightly less than that of endogenous talin1 in WT ChoK1 cells, but more than the remaining endogenous talin1 expression in knockdown cells (*Figure 5—figure supplement 1*). Interestingly, cells expressing the talin1 R8vvv-mNG mutant contained many more NAs (*Figure 5a,b,f,g,k*) and less and smaller FCs and FAs (*Figure 5a,b,f,g,l,m*) than control cells expressing WT talin1. Furthermore, cells expressing the R8vvv variant of talin1 also showed less traction than cells rescued with WT talin1 (*Figure 5c,h,n*). Consequently, with less traction and more NAs, the edge protrusion and retraction velocity was also faster in cells expressing the talin1 R8vvv-mNG mutant (*Figure 5d,i*). Moreover, a lower fraction of NAs and FCs in R8vvv mutant cells grew in size to FAs than NAs in cells with WT talin1 rescue (*Figure 5e,j,o,p*). Together, these results demonstrate that talin R8vvv mutation restricts NAs from maturing into focal adhesions.

Despite the reduced NA maturation, ChoK1 cells expressing the talin1 R8vvv mutant exhibited some large FAs (*Figure 5a,b and m*). Accordingly, we asked if these FAs could reflect a population of NAs that mature under recruitment of residual endogenous talin1 (*Figure 5—figure supplement 1*), or perhaps talin2, which we did not specifically knockdown. Western Blot analysis showed that talin2 expression was minimal in ChoK1 cells compared to talin1 and knocking down talin1 did not result in compensatory expression of talin2 (*Figure 5—figure supplement 2*). The traction magnitude of ChoK1 talin1 knockdown cells was significantly lower than of WT cells ectopically expressing talin1 or knockdown cells rescued with talin1 (*Figure 5—figure supplement 3*). Together, these data suggest that influence of a residual talin1 pool or of compensatory expression of talin2 on the behavior of ChoK1 cells with a talin1 knockdown and rescue by either R8vvv- or WT talin1 is minimal.

To ensure that the observed R8vvv NA maturation defect is independent of endogenous talin1 and talin2 expression, we lentivirally introduced the WT and the R8vvv variants of talin1 into talin1/2 null inner medullary collecting duct (IMCD) cells (see Materials and methods) (*Mathew et al., 2017*). Demonstrating the importance of talin for cell adhesion, talin1/2-null IMCD cells grew in suspension, and only adhered to the substrate when rescued with talin1 constructs. IMCD cells rescued with R8vvv-talin1 showed an adhesion phenotype that is very similar to ChoK1 rescued with R8vvv-talin1. Specifically, cells rescued with talin1 R8vvv had more NAs, smaller FCs and FAs, less traction, less

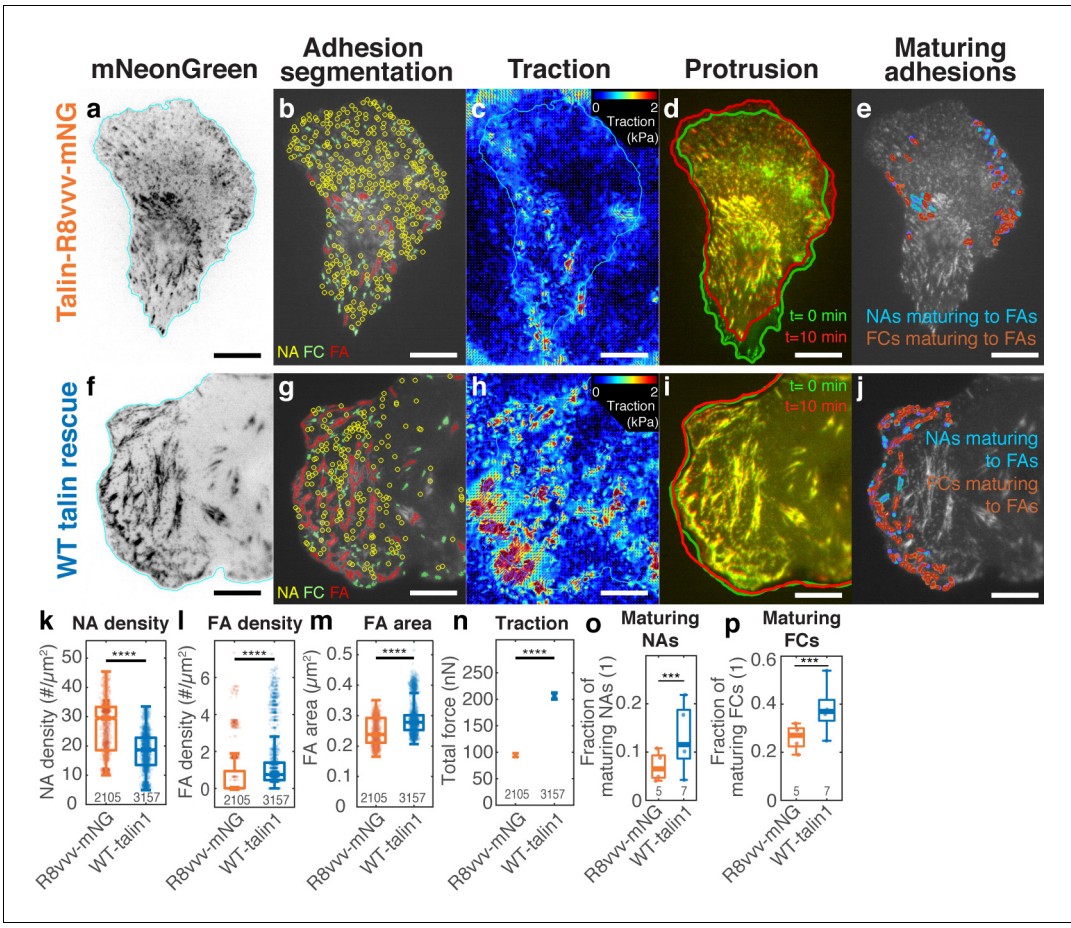

**Figure 5.** Expression of the talin1 R8vvv mutant in ChoK1 cells with endogenous talin1 knocked down results in the formation of denser NAs, but lesser and smaller FAs, lower traction, more active protrusions, and less maturing adhesions compared to WT talin. (a–j) Adhesion, traction, and protrusion phenotype of a representative ChoK1 cell on 5 kPa substrate expressing (a–e) talin R8vvv mutant or (f–j) WT talin. (a and f) inverted talin-mNeonGreen images. (b and g) detection of NAs, FCs and FAs. (c and h) traction force maps. (d and i) snapshots of computer vision-extracted cell boundaries at 0 and 10 min of a movie. (e and j) overlay of NAs and FCs that mature to FAs. (k–p) Box plots of (k) NA density, (l) FA density, (m) FA area, (n) total traction integrated over cell area, (o) fraction of NAs maturing to FAs relative to all NAs, (p) and of the fraction of FCs maturing to FAs (relative to all FCs). Number of adhesions imaged are listed under each box plot. Number of independently imaged cells for talin1 R8vvv-mNG and WT talin1-mNG rescue were 5 and 7, respectively. Scale bar: 10 μm. ****: p<1×10-30 by Mann-Whitney U test.

The online version of this article includes the following source data and figure supplement(s) for figure 5:

**Source data 1.** Source data for *Figure 5k*.
**Source data 2.** Source data for *Figure 5l*.
**Source data 3.** Source data for *Figure 5m*.
**Source data 4.** Source data for *Figure 5n*.
**Source data 5.** Source data for *Figure 5o*.
**Source data 6.** Source data for *Figure 5p*.
**Figure supplement 1.** Western blot for talin in WT ChoK1 cells, ChoK1 with shRNA-mediated knockdown of talin, WT cells with ectopic expression of talin1-mNG, WT cells with ectopic expression of talin1-R8vvv-mNG, and ChoK1 cells with shRNA-mediated knockdown of talin and ectopic expression of talin1-R8vvv-mNG.
**Figure supplement 2.** Western blot of talin1 and talin 2 in WT and knockdown (shRNA) cells.
**Figure supplement 3.** Traction forces generated by ChoK1 cells expressing the talin1 shRNA are smaller than GFP-talin1 and mNG-talin1 rescue cells.
**Figure supplement 4.** Expression of the talin1 R8vvv mutant results in formation of denser NAs, smaller FAs, lower traction, and less maturing adhesions compared to expression of the talin WT.

NAs and FCs maturing to FAs than IMCD cells rescued with WT talin1 (*Figure 5—figure supplement 4*). Thus, we concluded that the remaining large FAs likely result from talin1-vinculin complexes, which still form in the presence of the R8vvv mutation, albeit to a lesser extent (see *Figure 4C*).

## R8vvv mutation promotes talin recruitment overall but impedes traction growth rate

To investigate whether force-free complex formation of talin1 with vinculin affects talin recruitment itself, we compared the time of talin1 recruitment in R8vvv and WT talin1 rescue cells for non-maturing (G1) and maturing (G2) NAs. Consistent with the data in *Figure 2*, in both cases non-maturing NAs showed talin recruitment, on average, ~18–14 s prior to the initial rise in traction (*Figure 6a,c,e, g,i*), while maturing NAs showed a near-immediate, that is, with ~6 s before traction, talin recruitment (*Figure 6b,d,f,h,i*). This indicates that the ability of talin to bind vinculin in the absence of the force does not affect talin recruitment. For both WT and R8vvv mutant talin, the assembly rates were statistically indistinguishable between non-maturing and maturing NAs (*Figure 6j*). The rate of traction development in NAs, however, was significantly affected in talin1 R8vvv-mNG mutant cells. Overall, the traction growth rate was reduced in R8vvv cells, both for non-maturing and maturing NAs (*Figure 6k*). Moreover, whereas maturing NAs in WT talin1 rescue cells showed faster traction growth than non-maturing NAs, consistent with the data in *Figure 3c*, maturing adhesions in talin1 R8vvv-mNG mutant cells exhibited a force growth similar to that of non-maturing NAs (*Figure 6k*). Interestingly, the talin1 assembly rate of both NA types in R8vvv expressing cells was higher than WT talin1 rescue (*Figure 6j*). Inspection of individual NA trajectories (one example in *Figure 6—figure supplement 1a,b*) showed that R8vvv-talin is recruited fast but also disassembles fast. We speculate that this fast assembly relates to the high NA density shown in R8vvv-expressing cells (*Figure 5k*). These transient R8vvv-talin NAs also showed insignificant vinculin association (*Figure 6—figure supplement 1c,d*). These results suggest that the R8vvv mutation elevates talin's own recruitment rate in both maturing and non-maturing NAs, but interferes with force transduction in maturing NAs.

## Differential vinculin recruitment between non-maturing vs. maturing NAs vanishes with talin1 R8vvv mutation

Vinculin recruitment to the NA is critical for both force growth and adhesion maturation (*Figure 3*; *Thievessen et al., 2013*). To examine whether the assembly rate of vinculin is affected by talin's ability to form a complex with vinculin without initial tension, we performed two-channel imaging of vinculin-SnapTag-TMR-Star and WT- or R8vvv-talin-mNeonGreen (see Materials and methods). As performed previously, we captured and analyzed time-series of each pair of talin-vinculin signals in non-maturing vs. maturing NAs (*Figure 7a–h*) and quantified the vinculin assembly rate within 30 s after first detection (*Figure 7i*). In talin1 R8vvv-mNG mutant cells, vinculin assembly rates were statistically indistinguishable between non-maturing and maturing NAs, whereas in wild-type talin1-rescue cells vinculin rates were significantly higher in maturing NAs, consistent with the data acquired in control cells (*Figure 3a*). This result suggests that early vinculin binding to talin R8 domain indeed contributes to faster recruitment of additional vinculin. The insignificant difference in vinculin recruitment in R8vvv mutant cells for non-maturing vs. maturing NAs might be related to the reverted traction growth rates between the two NA groups observed in these mutant cells (*Figure 6k*). It is also worth noting that the vinculin signal in maturing NAs of cells with WT talin-rescue tended to keep increasing while talin intensity was relatively flat (t = 200 ~ 600 s in *Figure 7h,d*), suggesting that the number of exposed talin VBSs is increasing, and thus the number of bound vinculin proteins, over time under tension. The same trend was observed in talin1 R8vvv mutant cells (*Figure 7b,f*), but the vinculin recruitment rate was again much less than those found in WT talin1-rescue. Altogether, this data strongly suggests that vinculin recruitment is significantly reduced in the absence of tension-free vinculin-talin pre-complexation.

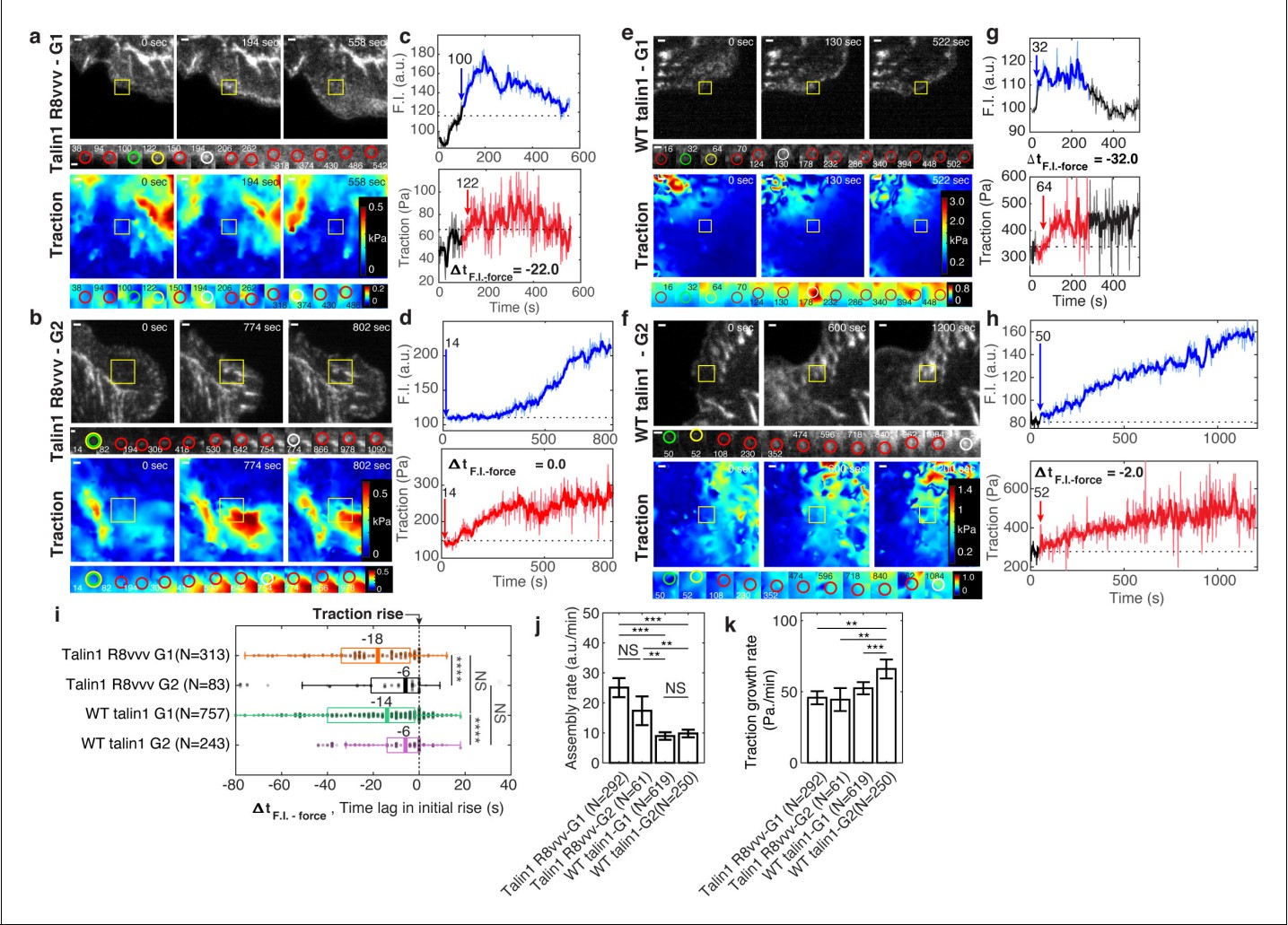

**Figure 6.** Expression of talin1 R8vvv-mNG mutant does not change the recruitment timing of talin to NAs, but reduces the force growth rate in NAs. (a–h) Representative talin (top) and traction force (bottom) images of talin1 R8vvv-mNG expressing cells (a–d) and WT talin-mNG rescue cells (e–h) within non-maturing (a,c,e,g) and maturing (b,d,f,h) NAs. (a,b,e,f) talin-mNG images (top) and traction images (bottom) of three different time points, that is at initial nucleation, at maximum fluorescence intensity, and at the end of the NA portion of the track. Yellow boxes indicate positions of example adhesions whose fluorescent signals and traction levels are shown in time lapse montages with finer resolution underneath. Green circles indicate the time points of initial talin signal rise, yellow circles show the time points of initial traction rise, and white circles show the time of the peak amplitude. Red circles show normal default detections without special events. Scale bar: 1 μm. (c–d, g–h) Traces of talin-mNeonGreen fluorescence intensity (top) and traction (bottom). Phases of the traces with significant fluorescence above background are indicated in blue and red, respectively. The black time series outside the colored signal are the background-subtracted intensities read at the first or last position detected by the particle tracker. Blue and red arrows mark the time points of the first intensity increase and the first traction increase, respectively (i–k) Distributions of time lags of fluorescence intensity onset relative to traction onset (i), talin assembly rates (j), and traction growth rates (k) of non-maturing (G1) and maturing (G2) NAs in talin1 R8vvv-mNG mutant and WT talin1-mNG rescue cells. Time integration time for calculating slopes in j and k was 20 and 60 s, respectively. *p<1×10$^{-2}$, **p<1×10$^{-3}$, ***p<1×10$^{-10}$, ****p<1×10$^{-30}$ by Mann-Whitney U test.

The online version of this article includes the following source data and figure supplement(s) for figure 6:

**Source data 1.** Source data for *Figure 6i*.
**Source data 2.** Source data for *Figure 6j*.
**Source data 3.** Source data for *Figure 6k*.
**Figure supplement 1.** An example of a nascent adhesion in an R8vvv-expressing ChoK1 cell exhibiting high assembly rate.

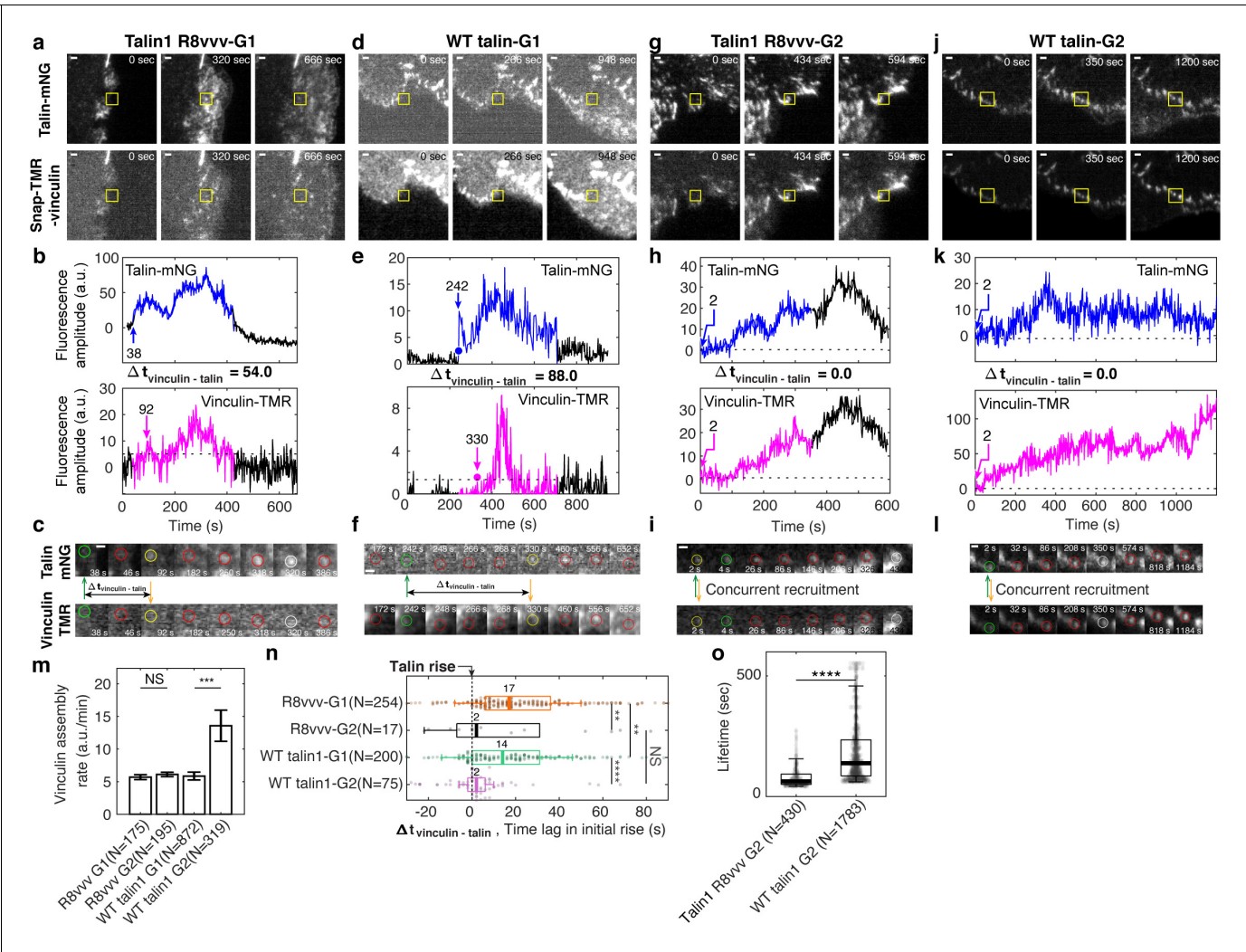

**Figure 7.** Vinculin recruitment is reduced in talin1 R8vvv mutant cells. (a,d,g,j) Representative two-channel time-lapse images of talin-mNeonGreen (top) and vinculin-SnapTag-TMR-Star (bottom) of G1 NA in a Talin1 R8vvv mutant cell (a), G1 NA in WT talin1 rescue cell (d), G2 adhesion in a Talin1 R8vvv mutant cell (g), and G2 adhesion in WT talin1 rescue cell (j). NAs of interest are indicated with a yellow box. Scale bar: 1 μm. (b–k) Time series of talin-mNeonGreen amplitude (top) and vinculin-SnapTag-TMR-Star amplitude (bottom) of G1 non-maturing (b,e) and G2 maturing (h,k) NAs in cells expressing the talin1 R8vvv mutant (b,h) and WT talin (e,k) constructs. Colored time periods (blue for talin, magenta for vinculin) indicate the phases where the adhesion is detected as a significant particle of robust trackability. The black time series outside the colored signal are the background-subtracted intensities read at the first or last position detected by the particle tracker. Blue and magenta arrows and the text around them indicate the time of talin and vinculin recruitment onset, respectively. (c,f,k,l) Time lapse montages of individual NAs shown in a, d, g, and j, respectively, overlaid with colored circles as detected centers of NAs of interest. Green circle represents the time point of initial talin signal rise, yellow the time point of initial vinculin signal onset, white the time of the peak amplitude, while red circles show normal default detections without special events. Talin and vinculin's initial recruitments are indicated with green and yellow arrows to highlight the time delay occurring between talin and vinculin in G1 adhesions and the concurrent recruitment in G2 adhesions, regardless of R8vvv mutations. (m) Vinculin assembly rates at non-maturing and maturing NAs in R8vvv mutant and WT talin rescue cells, quantified by the slope of vinculin-SnapTag-TMR-Star fluorescence intensity over the initial 20 s after the first detection in the talin-mNeonGreen channel. (n) Time delays of vinculin recruitment onset relative to talin recruitment onset of non-maturing vs. maturing NAs in talin1 R8vvv-mNG mutant and WT talin1 mNG cells. Vinculin recruitment onsets in non-maturing NAs are positive, that is, vinculin recruitment starts after talin. In contrast, vinculin recruitment onsets in maturing NAs are nearly coincidental with talin. See the text for further description. (o) Lifetimes of maturing NAs classified in talin1 R8vvv mutant and WT talin1 mNG rescue cells. ****$p < 1 \times 10^{-15}$, **$p < 0.05$ by Mann-Whitney U test. The numbers of adhesions (N), extracted from seven cells each for cells with talin1 R8vvv-mNG and WT talin1-mNG, are shown per each condition name at each panel.

The online version of this article includes the following source data for figure 7:

**Source data 1.** Source data for *Figure 7m*.
**Source data 2.** Source data for *Figure 7n*.
**Source data 3.** Source data for *Figure 7o*.

## Simultaneous talin-vinculin imaging confirms vinculin's recruitment after talin for non-maturing NAs and concurrent recruitment for maturing NAs

To confirm the recruitment order of talin and vinculin with respect to traction force development (*Figure 2*), we quantified the time difference between the first significant increase in talin fluorescence intensity and the first significant increase in vinculin fluorescence intensity (blue and magenta arrows in *Figure 7a–l,n*). For non-maturing NAs, both in talin1 R8vvv mutant and WT talin1-rescue cells, vinculin was delayed to talin (*Figure 7n*), consistent with the delay we inferred indirectly based on alignment of the fluorescent intensity increases with the first significant traction force increase (*Figure 2n*). In maturing NAs, vinculin and talin recruitment coincided (*Figure 7i,l,n*), also consistent with the indirect inference presented in *Figure 2o*. This shows directly that tension-free talin-vinculin pre-complexation enhances the probability of NA maturation. In more detail, vinculin recruitment in non-maturing NAs of talin1 R8vvv mutant cells was on average ~17 s after talin recruitment, whereas vinculin recruitment in the wild-type talin rescue condition was preceded by the talin recruitment on average by ~14 s (*Figure 7n*). We interpret the small but significant difference between the two recruitment time distributions as the result of the mutation in talin's R8 domain, which reduces the ability of vinculin to bind talin prior to mechanical unfolding. Moreover, even though some maturing NAs eventually grow also in talin1 R8vvv mutant cells, the absence of efficient vinculin binding to the VBS in R8 propagates into an overall less efficient vinculin recruitment. In agreement with this interpretation, we found that the lifetimes of maturing NAs in the mutant cells were much shorter than those in cells with WT talin1 rescue (*Figure 7o*). In summary, our data establishes that early association of talin with vinculin via the talin R8 domain is critical for accelerated vinculin binding, which in turn contributes to the development of the level of force transmission required for NA maturation.

## Discussion

Our experiments show that the maturation of NAs depends upon the concurrent early recruitment of talin and vinculin without tension. Previous models have inferred that tension across talin, which can establish direct bridges between integrins and actin filaments, is sufficient to trigger a conformational change that exposes several vinculin binding sites. These binding sites were thought to promote the recruitment of vinculin to further strengthen the link between the integrin-talin complex and F-actin (*Goult et al., 2018*; *Sun et al., 2016*). However, these models were derived primarily from observations in focal adhesions, that is, at a late stage of the maturation process (*Thievessen et al., 2013*; *Atherton et al., 2015*). Here, we exploit our ability to simultaneously measure nanonewton-scale traction forces and molecular associations within individual NAs using total internal reflection microscopy on high-refractive index soft substrates (*Gutierrez et al., 2011*; *Han et al., 2015*). Our data suggests that the tension born by an individual talin bridge between an integrin and actin filaments is insufficient to maximize the number of VBSs available for reinforcing the link with F-actin. This further lowers the lifetimes of catch-bond-like molecular associations (*Huang et al., 2017*; *Sun et al., 2016*; *Hákonardóttir et al., 2015*) between talin and vinculin, vinculin and actin, and talin and actin, resulting in turnover of the NAs (*Figure 8*, top). In contrast, the concurrent recruitment of talin and vinculin, perhaps as a preexisting cytosolic complex, immediately establishes a strong link between integrins and F-actin, as indicated by the immediate onset of traction forces. The fast loading rate promotes an efficient unfolding of the talin rod domain, which exposes several additional VBSs for further recruitment of vinculin and strengthening of the talin/F-actin interaction. This results in robust increase of traction force transmission and stabilization of catch-bond-like molecular associations that contribute to the maturation of the NA (*Figure 8*, bottom).

Our data also show that the colocalization of talin and vinculin is promoted by talin's R8 domain, which contains a VBS that is exposed for vinculin recruitment without tension-mediated unfolding of talin. We generated a talin mutant with a more stable R8 domain that reduces the spontaneous association with vinculin. Cells expressing this mutant have a large fraction of NAs that cannot mature into FAs and transmit only low-level forces. Ultimately, our data suggests that maturing NAs are likely to be initiated by an R8-mediated talin-vinculin association. Intriguingly, complex formation prior to binding integrin receptors requires spontaneous encounters of mobile talin and vinculin at

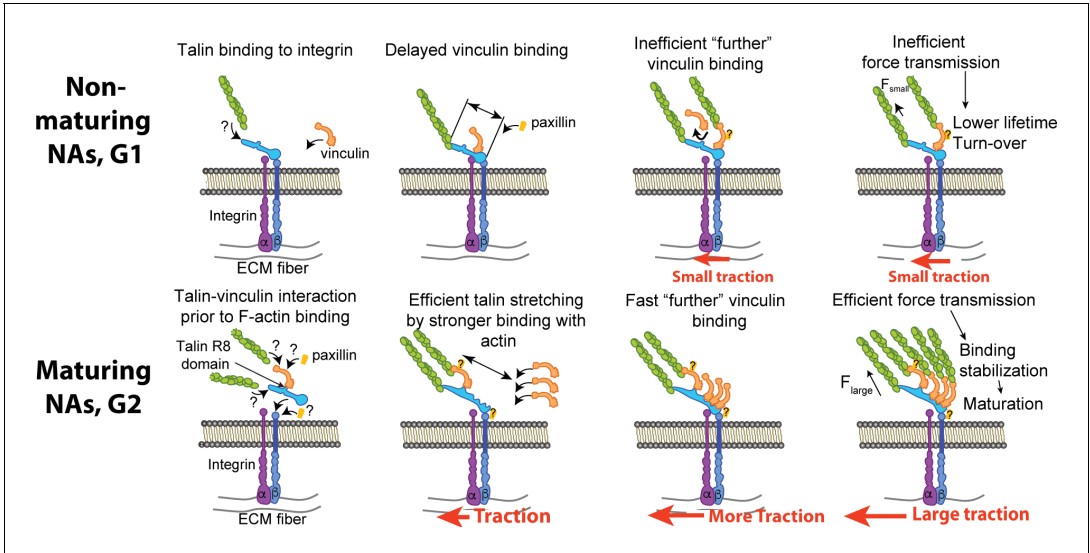

**Figure 8.** A suggested mechanism of differential recruitment of talin and vinculin determining maturation of nascent adhesions. (Top) For non-maturing NAs, talin binds to integrin before vinculin recruitment. Talin stretching might be limited to a shorter level, which limits the exposure of vinculin-binding-sites. Inefficient vinculin binding, in turn, limits the number of F-actin that can connect to the adhesion complex, allowing for only a low amount of tension across the complex. Insufficient loading level reduces the lifetime of catch-bond like associations between molecules, resulting in turnover of the NA complex. (Bottom) For maturing NAs, talin and vinculin interact before engagement with integrin. Upon concurrent recruitment to the NA traction force builds immediately. Talin might be stretched in a faster manner by pre-associated vinculin and talin's own binding to F-actin accommodate faster, efficient recruitment of additional vinculin. High loading levels across the complex stabilizes molecular bonds, which facilitates the maturation of the NA. The sites for paxillin binding, for example, to vinculin or β-integrin via FAK, are inferred from the literature (*Humphries et al., 2007*; *Turner et al., 1990*; *Hu et al., 2014*; *Lawson et al., 2012*).

the plasma membrane or even within the cytosol. These are likely rare events, which may explain the surprising finding that the number of maturing (G2) NAs (3.5 ± 1.6%, Mean ± Standard Error of the Mean, N = 20 movies) is low compared to G1 NAs (28.8 ± 3.5%, Mean ± S.E.M., N = 20 movies) among all NAs.

Despite the general phenotype of less NA maturation, talin1-R8vvv-expressing cells also exhibited a small number of large FAs (*Figure 5j*). These FAs are not due to residual endogenous talin1 or talin2 expression, as they are also evident in talin1/2 double knockout cells. Instead, these FAs in R8vvv-expressing cells likely result from one or more mechanisms. For example, (1) talin-R8vvv binding to vinculin is not abolished, only reduced (*Figure 4c*), (2) talin-R8vvv's association with integrins and F-actin could be quick enough to expose other VBSs, or perhaps (3) arise from one of the other NA classes (we focused largely on G1 and G2 here). Nonetheless, by performing three-color imaging of talin, vinculin, and traction forces (*Figure 7*), we show that R8vvv cells do have maturing NAs, but less of them, with reduced lifetimes, and importantly, with impaired vinculin association rates (*Figure 7m*). Thus, it appears that the second possibility might be less likely. Alternatively, we speculate that there may be a compensation effect at play where the fewer maturing NAs accumulate excess talin (*Figure 6j*).

Our findings regarding the talin R8 domain offer an alternative perspective on talin-vinculin association, including our own report describing talin's R3 domain as the weakest region that can unfold under tension (*Yao et al., 2016*; *Atherton et al., 2015*). Likewise, there are recent results that seem conflicting with one another. Using truncated talin fragments, it would appear that the talin R4-R8 fragment does not bind the vinculin head domain, whereas the talin R1-R3 fragment does (*Dedden et al., 2019*). However, another recent study has shown that a talin lacking R2R3 is able to interact with both inactive and active vinculin (*Atherton et al., 2020*). Moreover, a few studies have supported an idea that R3 requires force, albeit small, for engagement with vinculin (*Austen et al., 2015*; *Yao et al., 2016*; *Elosegui-Artola et al., 2016*): First, a study with a talin tension sensor has shown that talin1's engagement with the cytoskeleton must precede vinculin's binding to the N-terminal VBS on talin1 (*Austen et al., 2015*). Second, IVVI mutant in talin R3 has been reported to

prevent mechanotransduction, as assessed by traction exertion and YAP nuclear localization (*Elosegui-Artola et al., 2016*). Additionally, based on published data (*Goult et al., 2013*; *Lee et al., 2013*), the R3 domain is likely to be bound to the Rap1-interacting adaptor molecule (RIAM) prior to the force (*Vigouroux et al., 2020*). Thus, early, tension-independent interactions between talin and vinculin is more likely mediated by a different site, for which our data suggests the talin R8 domain.

Our finding of a role for R8-mediated talin-vinculin complex formation in the earliest stages of adhesion assembly is also somewhat unexpected in view of the paradigm that both full-length talin and vinculin reside largely in an auto-inhibited conformation that prevents mutual interaction. Indeed, a recent study showed that full-length talin1 in a closed conformation does not form a complex with vinculin regardless of vinculin's conformation (*Dedden et al., 2019*). In contrast, another study reported that only the 'activated' conformation of vinculin can form a complex with talin, or vice versa (i.e. an activated talin can form a complex with vinculin in its closed form) (*Atherton et al., 2020*). We speculate that both talin and vinculin are highly dynamic proteins that exist in an equilibrium that transitions between closed, open, and intermediate states in living cells (*Dedden et al., 2019*), and the potential activation energy necessary for activation could easily arise from F-actin-driven forces. Additionally, although biochemically not yet confirmed, a talin-vinculin precomplex appears feasible as the structure of full-length talin shows an exposed R8 VBS. These considerations collectively align with the observation that the occurrence of maturing G2 adhesions is less than the occurrence of non-maturing G1 adhesions. Thus, the conditions for efficient maturation, including the formation of a talin-vinculin complex without tension, are rarely fulfilled. Nonetheless, a rare but significant number of talin-vinculin precomplexes are sufficiently available to nucleate a population of G2 adhesions, which ultimately drives adhesion maturation.

In the case of non-maturing NAs, talin and vinculin were recruited significantly before our measurements could detect a significant traction onset. This finding implies that there are sub-populations of interacting talin and vinculin that transduce little force. Additionally, talin is present for a longer time than vinculin before traction onset in non-maturing adhesions. While this is conceptually consistent with a previous finding that vinculin binding to talin requires talin's actin binding for tension development (*Austen et al., 2015*), the time lag between talin and vinculin recruitment suggests that talin's sole engagement with F-actin without vinculin potentially impedes talin's own role as an integrin activator (*Shattil et al., 2010*) and promoter of integrin clustering (*Saltel et al., 2009*). Additionally, before vinculin binding, talin may be bound to RIAM (*Lee et al., 2009*), which is replaced by vinculin only after the R2R3 domain unfolds (*Goult et al., 2013*).

How a potential talin-vinculin precomplex promotes faster tension development and talin unfolding in maturing adhesions remains to be determined. Potential mechanisms imply that (1) the complex is also pre-bound to F-actin through the vinculin tail (as vinculin bound to talin is almost certainly in an open conformation) (*Humphries et al., 2007*; *Golji and Mofrad, 2013*) and (2) that the talin-vinculin interactions via talin's R8-domain do not interfere with talin's direct binding to F-actin, thus accelerating talin's actin-binding rate.

In maturing adhesions, paxillin is recruited concurrently with talin and vinculin. Which molecular partners in NAs and binding sites within those molecules are responsible for paxillin's concurrent binding needs further investigation. A previous study has shown that talin-vinculin interaction can facilitate efficient paxillin recruitment regardless of the paxillin-binding-site in vinculin's tail domain (*Humphries et al., 2007*). This finding suggests that paxillin's concurrent recruitment in maturing nascent adhesions is associated with the talin-vinculin precomplex but not necessarily via paxillin's direct binding to vinculin (*Carisey and Ballestrem, 2011*).

How non-maturing (G1) NAs switch to disassembly also necessitates further investigation. One potential scenario includes that talin is competitively bound by RIAM or DLC1 before it can associate with vinculin. Alternatively, vinculin binding to other VBSs within talin could interfere with force transduction and adhesion maturation (e.g. vinculin binding to R3, after partial unfolding by F-actin-mediated forces at ABS3). Whether vinculin's binding to such VBSs, that is, other than R8, interfere or synergize vinculin binding remains to be determined. However, our data suggests that when talin binds F-actin in G1 NAs – directly or indirectly via vinculin – the further recruitment of vinculin is much slower than that of G2 NAs where talin and vinculin arrive simultaneously in what we hypothesize is a precomplex.

Our data also indicates that the onset of traction force is accompanied by paxillin recruitment, regardless of the fate of the NA (*Figure 2*). This suggests that paxillin is recruited after vinculin, and

this is particularly true in non-maturing NAs. Indeed, the tension dependency of paxillin recruitment is well-established (*Schiller et al., 2011*). Our data suggests that tension-dependent recruitment of paxillin is through vinculin, which is consistent with previous findings that suggest paxillin recruitment can be induced by vinculin (*Humphries et al., 2007*) as it binds to the tail-domain of vinculin (*Turner et al., 1990*). Alternatively, paxillin has been reported to be recruited after focal adhesion kinase (FAK) in endothelial cells (*Hu et al., 2014*). Given evidence that talin can also be recruited by FAK (*Lawson et al., 2012*), paxillin's recruitment after talin and vinculin might be coincident with vinculin-paxillin binding mediated by FAK. In line with our measurements, a FRET-based tension sensor study showed that of the three molecules (paxillin, talin, and vinculin), paxillin levels correlate strongest with traction force levels (*Morimatsu et al., 2015*). Altogether, our findings agree with previous evidence that paxillin levels are an accurate reporter of traction levels, but not NA nucleation.

In summary, our work establishes an unexpected role for an early association of talin and vinculin in a tension-free state as a mechanical prerequisite to the further recruitment of vinculin to talin, which is the foundation of adhesion maturation. While the possibility of talin-vinculin pre-complexation has been discussed in previous studies (*Pasapera et al., 2010*; *Bachir et al., 2014*; *Khan and Goult, 2019*), their function has remained obscure until now. Here, we show that these interactions are an essential step in adhesion assembly. How the pre-complexations are promoted, and whether they are regulated by cellular signals, are two of the critical questions to be addressed in future studies.

# Materials and methods

## Key resources table

| Reagent type (species) or resource | Designation | Source or reference | Identifiers | Additional information |
|---|---|---|---|---|
| *E. coli* | BL21(DE3) | Sigma-Aldrich | CMC0016 | Electrocompetent cells |
| Cell line (*Cricetulus griseus*) | CHO-K1 | ATCC | CCL-61 | Courtesy of Dr.Horwitz, Allen Institute |
| Cell line *Mus musculus* | IMCD Talin1/2 KO | This paper | | Courtesy of Dr. Zent, Vanderbilt University |
| Lentiviral expression vector | pLVX-shRNA1 | Clontech | | Lentiviral construct for stable expression of shRNA. |
| Lentiviral expression vector | pLVX-CMV-100 | Addgene | Catalog # 110718 | Lentiviral construct for stable expression of ectopic proteins. |
| Transient expression vector | pCDNA3.1(+) | ThermoFisher Scientific | Catalog # V79020 | DNA construct for transient expression of ectopic proteins. |
| Primary antibody | Anti-talin 1 | Abcam | Catalog # 71333 | WB 1:1000 |
| Primary antibody | Anti-talin 2 | Abcam | Catalog # 105458 | WB 1:1000 |
| Secondary antibody | Goat anti-Mouse IgG (H+L) Cross-Adsorbed-Perosidaxe Antibody | ThermoFisher Scientific | Catalog # G-21040 | WB 1:1000 |
| Secondary antibody | Goat anti-Rabbig IgG (H+L) Cross-Adsorbed-Peroxidase Antibody | ThermoFisher Scientific | Catalog # G-21234 | WB 1:1000 |
| Transient expression vector | pCDNA-mNG-Talin-shRNA-R8 | | | R8 mutant of Talin1 resistant to shRNA. |
| Transient expression vector | pCDNA-mNG-shRNA-Talin | | | WT Talin1 resistant to shRNA |
| Transient expression vector | pCDNA-mNG-Talin | | | WT Talin1 |
| Lentiviral expression vector | pLVXCMV100-mNG-Talin-shRNA-R8 | | | R8 mutant of Talin1 resistant to shRNA. |

*Continued on next page*

*Continued*

| Reagent type (species) or resource | Designation | Source or reference | Identifiers | Additional information |
|---|---|---|---|---|
| Lentiviral expression vector | pLVXCMV100-mNG-Talin-shRNA | | | WT Talin1 resistant to shRNA |
| Lentiviral expression vector | pLVXCMV100-mNG-Talin | | | WT Talin1 |
| Transient expression vector | Paxillin-GFP | | | Original construct from Horwitz Lab |
| Transient expression vector | Vinculin-GFP | | | Original construct from Horwitz Lab |
| Transient expression vector | Talin-GFP | | | Original construct from Horwitz Lab |
| Recombinant expression vector | pet151-Talin1-R7R8-WT (murine) | GeneArt | | Original construct from Goult Lab |
| Recombinant expression vector | pet151-Talin1-R7R8-VVV (murine) | GeneArt | | Original construct from Goult Lab |
| Recombinant expression vector | pet151-Vinculin-Vd1 (murine) | | | Original construct from Goult Lab |
| Synthetic peptide | RIAM-TBS1 peptide (RIAM_6_30-C) DIDQMFSTLLGEMDLLTQSLGVDT-C | GLBiochem | | |
| Synthetic peptide | DLC1 peptide (DLC1_465_489-C) IFPELDDILYHVKGMQRIVNQWSEK-C | GLBiochem | | |

## Cell line

ChoK1 cells were received from the Horwitz lab (UVA). The ChoK1 cell line was regularly tested for mycoplasma infection in the Horwitz lab. Upon transfer to the Danuser lab, they were again tested for mycoplasma infection, which were negative. IMCD talin1/2 cells were obtained from Roy Zent's lab at Vanderbilt University, where they were tested for authenticity and mycoplasma infection before shipment.

## Cell culture

ChoK1 cells were cultured in Dulbecco's Modified Eagle Medium (DMEM) with 4.5 g/L D-Glucose, L-Glutamine, and Sodium Pyruvate (Gibco, 11995–065) supplemented with 10% Fetal Bovine Serum (Equitech-Bio, Inc, SFBU30), 1% Anti-Anti (Gibco, 15240112), and 1% Non-Essential Amino Acids (Gibco, 11140076). For transfection, cells were plated in a 6-well plate at ~30% confluency and transfected the next day with 350 ng of fluorescent protein-, or SNAP-tagged adhesion marker, 1 µg of pBluescript (Stratagene) as non-specific DNA, 10 µL of Lipofectamine LTX (Gibco, 15338030), and 2 mL of reduced serum Opti-MEM (Gibco, 31985088) according to the manufacturer's directions. Four hours after adding the DNA-lipid mixture to the cells, the media was replaced with full DMEM media. Twenty-four hr later, cells were trypsinized, and enriched with flow cytometry for low-level GFP-positive cells. Of this pool, 50,000 cells were seeded on fibronectin-coated (see below) traction-force microscopy substrates in pH 7.4 HyQ-CCM1 media (GE Lifesciences, SH30058.03), supplemented with 1.2 g/L of sodium bicarbonate and 25 mM HEPES. *mGFP-talin1* (provided by N. Bate and D. Critchley), *paxillin-eGFP* (provided by I. Schneider), and *mGFP-vinculin* (provided by M. Humphries) were used for adhesion-TFM two-channel experiments.

## Knock-out and knock-down experiments

For knock-out experiments, Murine Inner Medullary Collecting Duct (IMCD) Talin1/2 knockout cells (the kind gift of Dr. Roy Zent, Vanderbilt University [*Mathew et al., 2017*]) were grown in DMEM/F-12 Dulbecco's Modified Eagle Medium/Nutrient Mixture F-12 with L-Glutamine, 15 mM HEPES (Gibco, 11330–032) 10% Fetal Bovine Serum (Equitech-Bio, Inc, SFBU30), and 1% Anti-Anti (Gibco, 15240112) under standard cell culture conditions (5% $CO_2$, 37°C) and passaged regularly at ~70%

confluency. Owing to the complete knockout of Talin1/2, these cells are nonadherent and propagate in suspension. To rescue Talin1, IMCD cells were separately transduced using lentivirus with mNeon-Green-tagged WT and R8 mutant Talin1 (T1502V, T1542V, and T1562V) at low-levels using a truncated cytomegalovirus promoter (pLVX-CMV-100, https://www.addgene.org/110718/). To generate lentivirus, human embryonic kidney cells were transiently transfected with transfer vector (pLVX-CMV-100-mNG-Talin-18/pLVX-CMV-100-mNG-Talin-18-R8), viral packaging (psPAX2, https://www.addgene.org/12260/) vector, and viral envelope (pMD2.G, https://www.addgene.org/12259/) vector, at a 1:1:1 ratio (5 mg each) using 250 ml of Opti-MEM (Gibco, 31985088) and 45 ml (1 mg/ml) of polyethyleneimine. The final mixture was incubated for 15 min at room temperature and then transferred dropwise to human embryonic kidney cells. After 48 hr, the viral supernatant was gently removed from the human embryonic kidney cells, filtered with low protein binding 0.45 mm syringe filters, and added to IMCD Talin1/2 knockout cells at ~70% confluency with 12 micrograms/mL of polybrene. Twenty-four hr later the media was replaced with full DMEM/F-12 media, and changes in the adhesion behavior were observed 48 hr post-transduction. Approximately one-week post-infection, non-adherent cells were washed from the dish, and the remaining knockout cells were trypsinized and evaluated for traction force microscopy.

For knock-down experiments, a previously validated shRNA hairpin against talin (GGAAAGC TTTGGACTACTA), located on the N-terminus of talin1, was stably introduced into ChoK1 cells with a pLVX-shRNA1 lentiviral system (Clontech) and selected for with 5 µg/mL of puromycin. Western blot analysis indicated decreased levels of talin expression (*Figure 5—figure supplement 1*). For rescue experiments, talin1 from mouse *Mus musculus* was subcloned into the pCDNA3.1(+) mammalian expression vector (ThermoFisher Scientific, V79020). To make the reconstitution vectors insensitive to the shRNA, silent mutations were introduced into the corresponding shRNA target sequence. The new sequence of the reconstitution vectors is now GGAAGGCCCTAGACTACTA. This silent mutation was introduced into mNeonGreen talin1 pcDNA3.1 vector, which was used for WT reconstitution. The pCDNA3.1-mNG-Talin1-shRNA vector was mutated for talin1 R8vvv mutant with mutation sites at T1502V, T1542V, and T1562V. The R8 mutations (T1502V, T1542V, and T1562V, according to mouse numbering), alter the stability of the talin R8 domain. For vinculin imaging, mNeonGreen (Allele Biotechnology) was replaced with SNAP-Tag using seamless cloning, and labeled with SNAP-Cell TMR-Star (New England Biolabs, S9105S) or SNAP-Cell 647-SiR (New England Biolabs, S9102S) according to manufacturer's recommendations. All protein-coding regions of expression constructs were verified with traditional primer walking and Sanger sequencing.

## Expression of recombinant polypeptides

For in vitro analyses, murine vinculin Vd1 (residues 1–258), murine talin R7R8wt (residues 1357–1653), and R7R8vvv (residues 1357–1653; T1502V, T1542V and T1562V) were cloned into a pET151 vector (Invitrogen) and expressed in *E. coli* BL21(DE3) cells cultured in LB. Standard nickel-affinity chromatography was used to purify the His-tagged recombinant proteins as described previously (*Whitewood et al., 2018*). The proteins were further purified using anion exchange chromatography following cleavage of the 6xHis-tag with TEV protease. Protein concentrations were determined using their respective extinction coefficients at 280 nm.

## Circular dichroism (CD)

Spectroscopy was performed using a JASCO J-715 spectropolarimeter equipped with a PTC-423S temperature control unit. Denaturation profiles were measured from 20 to 80°C at 0.2°C intervals by monitoring the unfolding of α-helices at 208 nm. A total of 0.1 mg/mL of protein was dissolved in phosphate buffered saline (PBS). Measurements were made in a quartz cell of 0.1 cm path length.

## Fluorescence polarization assays

To determine if other binding partners of talin R8 domain except for vinculin can still bind to R7R8vvv fragment, the relative binding affinities were measured using an in vitro fluorescence polarization assay. The R8 interacting, LD-motif containing peptides from DLC1 and RIAM, that is, DLC1_465–489-C (IFPELDDILYHVKGMQRIVNQWSEK-C) and RIAM_6–30-C (DIDQMFSTL LGEMD LLTQSLGVDT-C), were coupled to a thiol-reactive fluorescein dye via the terminal cysteine. Peptides with a C-terminal cysteine were synthesized by GLBiochem (China). Stock solutions (i.e. peptide +

fluorescein) were made in phosphate-buffered saline (PBS; 137 mM NaCl, 27 mM KCl, 100 mM $Na_2HPO_4$, 18 mM $KH_2PO_4$, pH 7.4), 1 mM TCEP and 0.05% Triton X-100. Excess dye was removed using a PD-10 desalting column (GE Healthcare, Chicago, IL, USA). Titrations were performed in PBS using a constant 1 µM concentration of fluorescein-coupled peptide with increasing concentration of R7R8 fragment (either WT or vvv mutant); final volume 100 µM in a black 96-well plate. Fluorescence polarization (FP) measurements, in which the binding between the two polypeptides results in an increase in the fluorescence polarization signal, were recorded on a BMGLabTech CLARIOstar plate reader at room temperature and analyzed using GraphPad Prism. $K_d$ values were calculated with nonlinear curve fitting using a one-site total binding model.

## Analytical gel filtration

Gel filtration was performed using a Superdex-75 size exclusion chromatography column (GE Healthcare) at a flow rate of 0.7 mL/min at room temperature in 50 mM Tris pH 7.5, 150 mM NaCl, 2 mM DTT. A sample of 100 µL consisting of 100 µM of each protein was incubated at a 1:1 ratio at 25℃ for 10 min. The elution was monitored by a Malvern Viscotek SEC-MALS-9 (Malvern Panalytical, Malvern, UK).

## Western blot

Cells were transfected under identical conditions as they were for imaging experiments but with a 10 cm dish and sorted with a flow cytometer (FACS Aria II SORP) for low expression. Cells were lysed by adding 2x laemmli + 10% b-ME, vortexing, and heating at 95℃ for 10 min. Protein concentration was measured, and the same amount was loaded for each lane. The gel was semi-dry transferred with a turbo blot, and then incubated overnight in 5% milk in tris-buffered saline with 0.1% Tween 20 (TBST) at four degrees. Protein was visualized with an anti-talin antibody at 1:1000 and the loading control was visualized with anti-b-actin at 1:5000, each in 0.5% milk/TBST overnight at 4℃. Gels were then rinsed with TBST, and probed with IgG:horseradish peroxidase in 0.5% milk/TBST at 4℃ for 1 hr and then at room temperature for another 30 min. Gels were rinsed three times for 20 min in TBST and then detected with enhanced chemiluminescence.

## TFM substrate preparation

All silicone substrates had a diameter of 35 mm, a stiffness of 5 kPa (with the exception of *Figure 1— figure supplement 7*), were embedded with 580/605 or 640/647 ($\lambda_{EX}/\lambda_{EM}$) 40 nm-diameter beads, and were compatible with total internal reflection fluorescence illumination. Substrates were coated with fibronectin (Sigma Aldridge, F1141) the same day as imaging experiments were conducted by mixing 20 µL of a 10 mg/mL 1-ethyl-3-(3-dimethylaminopropyl) carbodiimide hydrochloride (EDC) solution, 30 µL of a 5 mg/mL fibronectin solution, and 2 mL of $Ca^{2+}$ and $Mg^{2+}$ containing Dulbecco's Phosphate Buffered Saline (DPBS, Gibco, 14040117) for 30 min at room temperature. Thereafter, the substrate was rinsed two times with DPBS, and incubated with 2 mL of 0.1% (w/v) bovine serum albumin in DPBS for another 30 min at room temperature, and rinsed several times with PBS prior to seeding with 50,000 transiently transfected cells.

## Live-cell TIRF imaging for TFM and adhesion proteins

After being seeded, the cells were allowed to adhere to the substrate for ~1 hr prior to imaging. This was one of the more effective ways to make sure that cells had active protrusions. Cells were imaged with a DeltaVision OMX SR (General Electric) equipped with ring-TIRF, which mitigates laser coherence effects and provides a more homogeneous illumination field. This microscope is equipped with a 60x, NA = 1.49, objective, and 3 sCMOS cameras, configured at a 95 MHz readout speed to further decrease readout noise. The acquired images were in 1024 × 1024 pixel format with an effective pixel size of 80 nm. Imaging was performed at 37℃, 5% carbon dioxide, and 70% humidity. To maintain the optimal focus, laser-based identification of the bottom of the substrate was performed prior to image acquisition, with a maximum number of iterations set to 10. Laser powers were decreased as much as possible, and the integration time set at 200 ms, to avoid phototoxicity. At the back pupil of the illumination objective, the laser power for both 488 and 568 nm lasers was ~44 µW. Imaging was performed at a frequency of 1 Hz for 5–10 min, and deviations between the alignment for each camera were corrected in a post-processing step that provides sub-pixel

accuracy. After imaging, cells were removed from the substrate with a 30% bleach solution, and the beads on the relaxed gel substrate were imaged for each cell position. Rapid imaging was necessary to mitigate swelling effects in the silicone substrate and to resolve traction forces in nascent adhesions.

## TFM force reconstruction

Bead images of the deformed gel – acquired when a cell was on the substrate – and a 'reference bead image' of the relaxed gel acquired after cell removal – were processed for traction reconstruction as described previously (*Han et al., 2015*). Briefly, the bead images of the deformed gel were compared with the reference image using particle image velocimetry. A template size of 17–21 pixels, and a maximum displacement of 10–80 pixels, depending on the bead density and overall deformation, were used for cross-correlation-based tracking of the individual beads. The displacement field, after outlier removal, was used for traction field estimation over an area of interest. The area of interest on the reference bead image was meshed with square grids of the same width, which depends on the average area per bead. The forward map, which defines the expected deformation of the gel at all bead locations given a unit force at a particular mesh of the force grid, was created by solving Boussinesq Eq. under the assumption of infinite gel depth. This forward map was then used to solve the inverse problem, that is given the measured field of bead displacements, the underlying traction field is determined. The solution to this inverse problem is ill-conditioned in that small perturbations in the measured displacement field can yield strong variation in the reconstructed traction field. To circumvent this problem, the traction field was estimated subject to L1-norm regularization. As discussed in detail in *Han et al., 2015*, L1-norm regularization preserved the sparsity and magnitude of the estimated traction field. Also as discussed and validated in *Han et al., 2015*, the application of L1-norm regularization over the L2-norm regularization most traction force microscopy studies employ is essential to resolve force variation at the length scale of the distances between individual nascent adhesions. The level of regularization is determined by a control parameter. We chose the parameter based on L-curve analysis, which guaranteed a fully automated and unbiased estimate of the traction field (*Han et al., 2015*). Strain energy, which represents the mechanical work a cell has put into the gel, was quantified as $1/2 \times$ displacement $\times$ traction, integrated over a segmented cell area. The unit of this integral is femto-Joule.

## Adhesion segmentation, detection, and tracking

Focal adhesions (FAs) and diffraction-limited NAs were detected and segmented as previously described (*Han et al., 2015*). Briefly, FAs from images of either labeled paxillin, talin, or vinculin were segmented with a threshold determined by a combination of Otsu's and Rosin's algorithms after image pre-processing with noise removal and background subtraction. Segmented areas larger than $0.2 \ \mu m^2$ were considered for focal contacts (FCs) or FAs, based on the criteria described by *Gardel et al., 2010*. Individual segmentations were assessed for the area and the length, which is estimated by the length of major axis in an ellipse that fit in each FA segmentation. FA density was calculated as the number of all segmentations divided by the cell area. Nascent adhesions were detected using the point source detection described in *Aguet et al., 2013*. Briefly, fluorescence images were filtered using the Laplacian of Gaussian filter and then local maxima were detected. Each local maximum was then fitted with an isotropic Gaussian function (standard deviation: 2.1 pixels, i.e. ~180 nm) and outliers were removed using a goodness of fit test (p=0.05). NA density was defined as the number of NAs divided by the entire cell area.

## Adhesion classification

From the adhesion tracks, features 1–9 in *Supplementary file 1A* were captured from the individual fluorescence intensity traces, and features 10–21 in *Supplementary file 1A* from the corresponding spatial properties, some in reference to the position and movement of the proximal cell edge and to the overlap with segmentations of focal adhesions and focal complexes. A total of nine class outputs were used to train the adhesion tracks. We believe that nine classes are a minimum number that represents a heterogeneity of the cell-ECM adhesions in a cell because the five were dedicated to classify NAs. The three classes assigned for FA classes can be viewed as insufficient in terms of

representing a whole FA population and can be expanded further into a larger number of classes, but they were sufficient in this study because NAs were its main scope.

The classification was accomplished using a cubic support vector machine (SVM), which was proven to be the most accurate among linear classifiers. The classifier was evolved in a human-in-the-loop fashion, i.e. the user iteratively adjusted machine-generated classifications. The initial training data was labeled with qualitative criteria described in *Supplementary file 1B*. To facilitate the labeling process, an automatic, filtering-based, labeling was also employed (see *Supplementary file 1C*).

Both manual labeling and automatic labeling have advantages and drawbacks in terms of classification accuracy: while the manual labeling is less consistent due to subjectivity and human error, the automatic labeling has deficiencies in terms of incompleteness of the filtering criteria in capturing all essential properties of different adhesion classes. To overcome these drawbacks, both methods were employed in a way that the automatic labeling was performed first, and then manual labeling was added for insufficiently-labeled classes. During the manual labeling, adhesion classifications were immediately updated and presented to the user to allow class reassignments of selected adhesions. The labeling process was regarded to be completed once at least 10 adhesions were labeled for each class. To remove classification bias due to potential imbalance in the number of labels across the classes, the minority classes were oversampled, and the majority classes were under-sampled, based on the mean number of labels (*Krawczyk, 2016*). After training a classifier on one movie, for a new movie, another iteration of automatic-and-manual labeling was executed to update the classifier, which was applied to predict the classes of adhesions in the movie. Separate classifiers were built for talin-, vinculin-, and paxillin-tagged adhesions.

## Statistical methods

Processed data such as traction, assembly rate, time delays, lifetimes, adhesion densities and areas in different conditions were compared using Mann-Whitney non-parametric test since all individual distributions do not follow a normal distribution. Testing for normal distribution was done by one-sample Kolmogorov-Smirnov test using a 'kstest' function in Matlab.

## Software availability

A GUI-based Matlab software is shared via GitHub at https://github.com/HanLab-BME-MTU/focalAdhesionPackage.git; *Han, 2021*; copy archived at swh:1:rev:6aeb3593a5fd3ace9b0663d1bf0334decfb99835.

## Acknowledgements

We thank Roy Zent (Vanderbilt) and Olga Martha Viquez (Vanderbilt) for providing IMCD talin 1/2 double KO cells. We also thank Joseph Chi and Dana Reed for preparing DNA constructs and assisting with western blots, and Assaf Zaritsky (Ben Gurion University) for providing helpful comments about machine learning. This work was funded by the following grants: NIH R15GM135806 (SJH) NIH F32GM117793 (KMD), NIH P01GM098412 (AG, ARH, and GD), NIH R35GM136428 (GD), Biotechnology and Biological Sciences Research Council grant BB/N007336/1 (BTG) and a Human Frontier Science Program grant RGP00001/2016 (BTG).

## Additional information

### Funding

| Funder | Grant reference number | Author |
|---|---|---|
| National Institutes of Health | R15GM135806 | Sangyoon J Han |
| National Institutes of Health | F32GM117793 | Kevin M Dean |
| National Institutes of Health | P01GM098412 | Alex Groisman<br>Alan Rick Horwitz<br>Gaudenz Danuser |
| National Institutes of Health | R35GM136428 | Gaudenz Danuser |

| Biotechnology and Biological Sciences Research Council | BB/N007336/1 | Ben Goult |
|---|---|---|
| Human Frontier Science Program | RGP00001/2016 | Ben Goult |

The funders had no role in study design, data collection and interpretation, or the decision to submit the work for publication.

## Author contributions

Sangyoon J Han, Conceptualization, Data curation, Software, Formal analysis, Funding acquisition, Validation, Investigation, Visualization, Methodology, Writing - original draft, Project administration, designed the experiments and performed TFM reconstruction, nascent adhesion analysis, and machine learning from the images; Evgenia V Azarova, Investigation, Methodology, cultured IMCD talin1/2 KO cells and expressed WT talin1 and talin1 R8vvv vectors and performed imaging experiments with talin construct and its variations (e.g. R8vvv mutant); Austin J Whitewood, Data curation, Validation, Investigation, Methodology, performed biochemical experiments and analyses; Alexia Bachir, Conceptualization, Investigation, performed imaging experiments for WT GFP-tagged vinculin and paxillin; Edgar Guttierrez, Resources; Alex Groisman, Resources, Provided high-refractive-index silicone substrates for traction force microscopy; Alan R Horwitz, Conceptualization, Data curation, Funding acquisition, Methodology, Project administration; Benjamin T Goult, Conceptualization, Resources, Data curation, Formal analysis, Investigation, Methodology, Writing - review and editing, provided suggestions for talin structure and mutation and performed biochemical experiments and analyses; Kevin M Dean, Data curation, Investigation, Methodology, Writing - review and editing, created ChoK1 cells with talin1 shRNA and talin1 R8vvv expression and performed imaging experiments with the cells; Gaudenz Danuser, Conceptualization, Supervision, Funding acquisition, Investigation, Methodology, Project administration, Writing - review and editing

## Author ORCIDs

Sangyoon J Han (iD) https://orcid.org/0000-0002-1384-665X
Evgenia V Azarova (iD) http://orcid.org/0000-0002-3846-9176
Benjamin T Goult (iD) https://orcid.org/0000-0002-3438-2807
Kevin M Dean (iD) https://orcid.org/0000-0003-0839-2320
Gaudenz Danuser (iD) https://orcid.org/0000-0001-8583-2014

## Decision letter and Author response

Decision letter https://doi.org/10.7554/eLife.66151.sa1
Author response https://doi.org/10.7554/eLife.66151.sa2

# Additional files

## Supplementary files

• Supplementary file 1. Tables for information used in machine learning of adhesions. (A) Features used for classification of adhesions. (B) Nine adhesion groups defined heuristically. (C) Automatic labeling criteria.

• Transparent reporting form

## Data availability

All sequencing data are freely available on Zenodo. A GUI-based Matlab software is shared via GitHub at https://github.com/HanLab-BMEMTU/focalAdhesionPackage.git (copy archived at https://archive.softwareheritage.org/swh:1:rev:6aeb3593a5fd3ace9b0663d1bf0334decfb99835/).

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
