## [Decision Letter]

**Acceptance summary:**

The manuscript nicely demonstrates how talin, vinculin and paxilin coordinate maturation of nascent cell-substrate adhesions and mechanotransduction to control motility and shape of cells.

**Decision letter after peer review:**

[Editors’ note: the authors submitted for reconsideration following the decision after peer review. What follows is the decision letter after the first round of review.]

Thank you for submitting your work entitled "Formation of talin-vinculin pre-complexes dictates maturation of nascent adhesions by accelerated vinculin recruitment" for consideration by *eLife*. Your article has been reviewed by three peer reviewers, including Reinhard Fässler as the Reviewing Editor and Reviewer #1, and the evaluation has been overseen by a Senior Editor. The following individuals involved in review of your submission have agreed to reveal their identity: Nico Strohmeyer (Reviewer #2); Thomas Weidemann (Reviewer #3).

The reviewers spent quite a bit of time reviewing and then discussing this paper, in part because of the numerous concerns that were raised in the reviews. Although the reviewers found the manuscript very interesting and considered the possibility of having you resubmit a revised manuscript after clarifying the concerns/comments, it was ultimately decided that the experiments that needed to be done were too many for the time frame of 2-3 months. Based on these discussions, we regret to inform you that your work will not be considered further for publication in *eLife*. We strongly encourage you, however, to submit a new version of your manuscript in case you address our concerns and comments.

Reviewer #1:

The paper reports five classes of NAs based on their different mechanical behavior and recruitment dynamics of talin, vinculin and paxillin. The authors focus their study on two NA types, non-maturing and maturing NAs, and demonstrate that they greatly differ with respect to their force levels and talin/vincluin recruitment. Whereas maturing NAs recruit talin preloaded with vinculin, non-maturing NAs a non-vinculin bound talin, which prevents them to transduce high forces to the integrin/ligand bond and strengthen the bond.

The described methodology (combination of machine learning, high-resolution traction force microscopy, single-particle-tracking and fluorescence fluctuation time-series analysis) and findings (division of NAs into different subgroups based on their kinetic and mechanical properties and the importance of a pre-formed talin-vinculin complex for NA maturation to FAs) are of interest for the readership of *eLife*. However, publication in *eLife* requires additional analysis:

1) To validate their results the authors should test with simple manipulations whether the balance between the different adhesion subclasses can be shifted (e.g. Mn^2+^-treatment, PLL, different substrate stiffness) to validate their findings.

2) It is not clear why the authors used ChoK1 cells as their "model system" and expressed fluorescently tagged talin, vinculin or paxillin on top of the endogenous proteins or after knock-down of talin1 instead of a clean knockout background. How can the authors be sure that the fluorescently tagged-proteins are incorporated into NAs and FAs similar to untagged proteins?

Regarding the talin1 knockdown: is it know if talin2 is expressed in ChoK1 cells? Did the authors observe differences in the expression levels of talin1 wt-mNG or talin1 R8vvv-mNG? Why is the mNG-tagged talin1 running at a seemingly similar MW then the endogenous talin1 in Figure 5—figure supplement 1?

3) Their data demonstrate that the talin1 R8vvv mutation, which prevents talin-vinculin pre-complex formation, restricts NAs from maturing into focal adhesions. Is there a way to increase talin-vinculin pre-complex formation (e.g. by expressing an activated form of vinculin) to increase the number of maturing (G2) NAs? To support their hypothesis of the importance of pre-assembled talin-vinculin complexes it would be interesting to test the effect of talin1 variants impaired in actin binding to determine the importance of these independent actin linkages. What has a stronger effect?

4) The authors write that their analysis suggests, "that a very large number of adhesion protein aggregates, detectable through either talin, vinculin, or paxillin recruitment, never engage with the substrate." Is it not an overstatement to say that the failure to detect traction forces shows that these adhesions are not engaged with the substrate? Could they engage with the substrate but not link to actin?

Reviewer #2:

Han et al. describe a pre-complex of talin and vinculin formed in the cytosol, in which the vinculin vd1 domain interacts with the talin R8 domain. The authors demonstrated this complex with recombinant protein fragments of talin and kindlin. An R8vvv mutation of talin mechanically stabilizes R8, reduces the formation of the pre-complex and the number of maturing NAs in cells. The authors used high resolution traction force microscopy and TIRF to conclude that the talin-vinculin pre-complex is essential to mature NAs and sustain traction forces applied to them characterized. While the data and the model potentially provide a functional role of R8 in NA maturation, several questions arise.

1) To better understand the experimental setup, a figure explaining the method as well as an image of the flexible substrate with embedded beads measuring traction forces (including the processing steps) would be helpful. Additionally, the bead density and average distance between beads in these substrates should be reported. The mechanical properties of the substrates should be evaluated, as they might differ from batch to batch. Technically, I am wondering where the different background traction levels (see for example Figure 2, different traction force traces) or the traction measured outside a cell (see Figure 5A-C) arise from.

2) Talin and vinculin are known to exhibit auto-inhibited conformations, in which an interaction between R8 and Vd1 may not be possible. Can the authors show in a more physiological setting (preferably full-length proteins) that talin and vinculin form a complex in solution? Additionally, in Figure 4C it seems that talin1 R8vvv can still bind vinculin, although not as proficient. Can the authors show the importance of vinculin binding to R8 for the maturation of NAs into FCs more directly by mutating only the vinculin binding site in R8?

3) The authors state that in maturing (G2) NAs vinculin and talin are recruited together. However, in Figure 2O it appears that in about 25% (and in Figure 7J ~50%) of cases, vinculin is recruited before talin. In Figure 7H it appears that vinculin is recruited to NAs 14 s earlier than talin. Could the authors elaborate how vinculin is recruited before talin is visible?

4) How do the authors exclude the possibility that talin is quickly unfolded (as described in the legend of Figure 8) by forces from the cytoskeleton, which induces a fast recruitment of vinculin to talin in maturing NAs (which may be dependent on R8)? This process could be faster than the sampling rate and hence a pre-complex is not required to form in the cytosol.

5) The authors show multiple traction force maps and images acquired for cells that show very different phenotypes although being the same cell line. In Figure 1A-F ChoK1 cells are shown that overexpress mGFP-tagged talin, vinculin or paxillin but have very different morphologies, sizes (d-f) and exhibit very different traction forces (a-c) where the scale is 0-300 Pa for talin, 0-1800 Pa for vinculin and 0-1000 Pa for paxillin. In Figure 2 it is similarly diverse where traction force scales range from 0-800 Pa (in h) to 0-4000 Pa (in a). What is the reason for this heterogeneity in morphologies and traction forces within the same cell line?

6) In maturing (G2) NAs, does the constantly increasing traction force arise from the accumulating more vinculin at the adhesion site (i.e. stronger connection to the actomyosin), or from the growth of the adhesion site (i.e. accumulating more integrins)? In Figure 2G, a continuous recruitment of talin is observed, which would argue for the increase in traction forces due to the recruitment of additional integrins. How would the authors explain the pre-formation of talin-vinculin complexes to be important for the recruitment of integrins to the adhesion site (in maturing NAs)?

7) For testing the effect of the R8vvv talin mutant in cells, the authors used shRNA to knock-down talin1 levels before rescuing cells with either WT or R8vvv talin1. How is the talin expression level of WT talin1 rescued cells compared to R8vvv rescued cells? Further, an assessment of the morphology (whole cells and possible adhesion sites) of talin1 KD cells would be beneficial to appreciate the difference between R8vvv and WT talin 1. This would exclude the possibility that apparent differences between R8vvv and WT talin1 re-expressing cells arise from potentially different talin1 expression levels, a potential expression of talin2 in response to reduced talin1 levels or residual expression of WT talin1 in KD cells.

8) The authors state that the talin-vinculin pre-complex formation is reduced in talin1-R8vvv expressing cells. However, talin1-R8vvv expressing cells show adhesion sites larger than NAs (Figure 5A). How do larger adhesion sites form in cells expressing talin1 R8vvv, if the vinculin binding to R8 is essential for NAs maturation? Representative image of labeled vinculin and talin within one cell (as analysed for Figure 7) would be beneficial to understand how the vinculin recruitment is affected by the R8vvv mutation.

9) In their very clear conclusive Figure 8 the authors depict different sites for paxillin in non-maturing vs. maturing NAs. Do the authors have experimental evidence for this? Further, the legend describes hypotheses, which can explain experimental data but experimental evidence from the reported data is limited. For example, that talin stretching is limited to a shorter level in non-maturing NAs and talin stretches faster in maturing NAs has, to my point of view, not been shown in the presented data.

Reviewer #3:

Han et al. describe in great detail the molecular decision process that leads from transient nascent adhesions (NAs) forming constantly at the plasma membrane to focal adhesions (FAs) stably connecting the ECM with the actin cytoskeleton. The assembly kinetics of three GFP-tagged NA components (talin, vinculin and paxilin) is tracked down by TIRF microscopy and their behavior is correlated with local traction force determined by imaging the displacement of fluorescent beads in the hydrogel support, on which the cells grow. The authors apply machine learning algorithms to identify two classes of events both linked to force-assisted growth but with opposite fates. Using a new talin mutant, it is shown that binding of vinculin at a specific talin domain (R8) is required to establish the feed forward control for further vinculin recruitment and the steady build-up of local traction. In contrast, sequential binding of talin, vinculin and paxilin leads to maturation abort. I consider the study to be compelling, the technology pioneering and the results significant. Given that the topic is quite complex, the manuscript is of remarkable clarity. A direct comparison with R3 talin-mutants would add further evidence, however, such experiments are probably not within the scope of a revision. My listed points represent mainly comprehension questions that could encourage improvements of the text. Overall, I am in full support of publication.

1) Is there statistics on how many G2-class NAs actually mature into FAs?

2) "Event-based time series analysis" measures the appearance of fluorescently tagged proteins. What is the sensitivity of the method? How many mGFPs have to accumulate to detect a spot?

3) At the stage of Figure 2, is the existence of an unstrained talin-vinculin pre-complex really a stringent conclusion? The time resolution of imaging is one second. During this time bin, proteins with a Kd in the μM range may undergo several binding and unbinding cycles. Likewise, the forces may exhibit fast fluctuations especially during the onset when only a small number of force transmitting proteins have been accumulated (see Figure 3B). Thus, there is plenty of "temporal" space for different scenarios of "what is first" that is technically inaccessible. Please comment.

4) Figure 3B, the average force growth in the first 10 seconds may simply be zero due to the sensitivity limit of TFM. Or does this represent a finding?

5) Are G1-NAs programmed to terminate, because talin has too much time for erroneous interactions? Explicitly, do traction-dependent vinculin binding site (VBS), other than R8, represent "wrong sites" in the sense that cooperative vinculin assembly is even prevented?

6) At which stage of NA maturation then must force be sensed?

7) Is mutant R8vvv-talin totally impaired in vinculin binding (Figure 4D) or is there residual probability for exposing the VBS (Figure 4C)? The biochemical result is contradictory.

8) The difference between vinculin epitopes in domain R3 and R8: As a naïve reader one would assume that a VBS of the relaxed state locates at the surface of the protein. I consider it conceptually difficult to acknowledge that the VBS in domain R8 is active in an unstrained state but nevertheless requires unlocking of the threonine belt and partial unfolding of the α-helical bundle; maybe this warrants a sentence or two.

9) Cells expressing the R8vvv mutant are clearly impaired in NA maturations as documented by the cellular analysis (Figure 5). However, there are mature FAs. This may either relate to residual wildtype talin that could not be suppressed by shRNA (as the WB shows, Figure 5—figure supplement 1), residual binding of the mutant R8vvv-talin or even an undefined pathway into the cooperative vinculin assembly. These possibilities are not mentioned.

10) The traction growth rates in Figure 6K reach much more into the negative range than those in Figure 3B and 3C. Do negative values signify initial forces that disappear? In addition, there is quite a large difference in the scale of y-axis in Figure 6K (100-200 Pa/sec) as compared to Figure 3B and C (1-2 Pa/sec).

11) The body text referring to Figure 6 at the end of the paragraph says "These results suggest that the vinculin-talin pre-complex is essential for the development of force across NAs…". In my opinion this statement still remains an assumption, since I could not find direct evidence in Figure 6. The behavior of the mutant R8vvv-talin, if G2 characteristics are fulfilled, appear to be indistinguishable from the mGFP-tagged version described in Figure 2. The fact that there is a problem with force growth can only be explained in the light of Figure 7. Re-ordering these arguments may be considered.

12) Again, the two expression systems lead to a different range of values: The absolute vinculin assembly rates in Figure 7I as well as the rate increase between G2 vs G1 are much smaller than in Figure 3A. Is this due to different protein expression levels? Why were different integration times used (30 seconds in Figure 7I in contrast to 10 seconds in Figure 3A)?

13) Is actin always engaged first and then the integrin tails? In other words, is there support for a "search mode" of a small number of competent actin-talin-vinculin-heads for integrin tails linked to the ECM (Figure 8, G2, first panel)?

[Editors’ note: further revisions were suggested prior to acceptance, as described below.]

Thank you for submitting your article "Talin-vinculin precomplex drives adhesion maturation by accelerated force transmission and vinculin recruitment" for consideration by *eLife*. Your article has been reviewed by three peer reviewers, including Reinhard Fässler as the Reviewing Editor and Reviewer #1, and the evaluation has been overseen by Jonathan Cooper as the Senior Editor. The following individuals involved in review of your submission have agreed to reveal their identity: Nico Strohmeyer (Reviewer #2); Thomas Weidemann (Reviewer #3).

Summary:

Cell adhesion to the extracellular matrix is mediated by integrins which require an activation step and exposure to tension for ligand binding, adhesion site formation and maturation. Han and co-workers elegantly demonstrate that the maturation of small into large adhesions depends on the isochronic association of talin (integrin activator) and vinculin (force transmitter) with integrins and the generation of increasing forces. The study represents an inspiring example for the power of microscopy and machine learning to approach the emergence of cellular functions that involve molecular decision making.

Essential revisions:

1) The conclusion that the concurrent recruitment of talin and vinculin to maturing NAs corresponds with the existence of a cytoplasmic precomplex of the two proteins. Although it is likely that such precomplexes form prior to NA recruitment, it has not been demonstrated in the current paper. Hence, it is necessary to tone this claim down in the title, Abstract, Results section and Discussion of the manuscript.

2) Use same assembly rates (Figure 3A, Figure 6J, and Figure 7M) in y-axis. It is not clear what is meant by 1/min in Figures 3A/7M, and what is the rational for the different time frames analysed (Figure 3A: 2 min; Figure 6: 1 min; Figure 7: 0.5 min).

3) Although the authors state that the R8vvv mutation does not affect the talin recruitment rate, Figure 6J shows that in both G1 and G2 NAs talin R8vvv is recruited at higher rates than WT talin1. Test the rates for statistical significance (for G1 and G2 R8vvv vs. WT). In case talin1 R8vvv is indeed recruited faster to adhesion sites, the authors should try finding an explanation.

4) The authors state that traction increase for talin1 R8vvv is reduced compared to WT talin1. The statistical test in Figure 6K to corroborate this statement is missing.

5) Subsection “Simultaneous talin-vinculin imaging confirms vinculin’s recruitment after talin for non-maturing NAs and concurrent recruitment for maturing NAs”: should probably reference Figure 7N (instead of Figure 7J) and Figure 7O (instead of Figure 7K).

6) "Fluorescence fluctuation analysis" may be misleading as it usually refers to FCS, PCH, i.e., methods that involve a statistical correlation analysis. Here the authors directly analyze the time traces by thresholds and slopes.

7) The density 2.2 beads/µm2 cannot be reproduced from average spacing 0.42 µm via the relation: density = 1/sqrt(spacing).

8) "Both peptides bound…" This sentence sounds strange and may be changed to "Both peptides showed comparable affinity towards wildtype R7R8 and the R8R7vvv mutant, confirming.…"

9) "The threonine belt in talin R8 is responsible…" The wording is not clear since only the binding epitope itself can be responsible for binding and the threonine belt is, rather indirectly, regulating accessibility and hence the on-rate. Do these data suggest that RIAM and DLC1 use binding epitopes spatially distinct from the VBS within the R8 bundle? Or is RIAM and vinculin binding indeed competitive?

---

## [Author Response]

[Editors’ note: the authors resubmitted a revised version of the paper for consideration. What follows is the authors’ response to the first round of review.]

Reviewer #1:The paper reports five classes of NAs based on their different mechanical behavior and recruitment dynamics of talin, vinculin and paxillin. The authors focus their study on two NA types, non-maturing and maturing NAs, and demonstrate that they greatly differ with respect to their force levels and talin/vincluin recruitment. Whereas maturing NAs recruit talin preloaded with vinculin, non-maturing NAs a non-vinculin bound talin, which prevents them to transduce high forces to the integrin/ligand bond and strengthen the bond.The described methodology (combination of machine learning, high-resolution traction force microscopy, single-particle-tracking and fluorescence fluctuation time-series analysis) and findings (division of NAs into different subgroups based on their kinetic and mechanical properties and the importance of a pre-formed talin-vinculin complex for NA maturation to FAs) are of interest for the readership of eLife. However, publication in eLife requires additional analysis:1) To validate their results the authors should test with simple manipulations whether the balance between the different adhesion subclasses can be shifted (e.g. Mn^2+^-treatment, PLL, different substrate stiffness) to validate their findings.

We thank the reviewer for this suggestion. To evaluate if a simple manipulation could shift the balance between adhesion classes, we compared adhesion classification for ChoK1 cells transfected with vinculin GFP on both 5 kPa and 18 kPa stiff substrates. As anticipated, the subpopulation for each group showed significant and meaningful differences between the two stiffness conditions. For example, we found major reduction in G1, G2, G3 and G5 groups – all NAs – and an increase in G7 and G8 groups – major FAs – for cells cultured on the stiffer 18 kPa substrates. We believe that these differences can be attributed to the increased stability of FAs in stiff environments.

2) It is not clear why the authors used ChoK1 cells as their "model system" and expressed fluorescently tagged talin, vinculin or paxillin on top of the endogenous proteins or after knock-down of talin1 instead of a clean knockout background. How can the authors be sure that the fluorescently tagged-proteins are incorporated into NAs and FAs similar to untagged proteins?

While there are pros and cons with any cell type, this study is a follow up on a series of papers by Rick Horwitz and colleagues, which had provided extensive (and widely accepted) insights into adhesion formation and maturation. Historically, Rick selected the CHO epithelial cell because its adhesion assembly process was slower than seen in highly contractile cells, and therefore the individual molecular steps of adhesion formation could be readily resolved by live cell imaging. Importantly, the literature suggests that the adhesion assembly and maturation process in CHO cells has been largely recapitulated in other cells types.

As requested, we now reproduce our key findings in talin1/2 double knockout IMCD cells. Of note, our concern with these cells is their potential for long-term genetic adaptation and phenotypic drift owing to the disruption of a central adhesion element. Indeed, these cells propagate in suspension, which is likely accompanied by significant shifts in gene expression and cell signaling. Technically, the IMCD cells were challenging to transiently transfect, which led us to stably transduce them with lentiviruses. Virus production was also very inefficient because full-length GFP-tagged talin1 is well-beyond the viral packaging capacity of most lentiviral systems. Nonetheless, after months of troubleshooting, we succeeded in the expression of both WT and R8vvv mutant talin in knockout cells. As in ChoK1 cells, rescue of IMCD cells expressing with R8vvv mutant showed significantly less adhesion maturation (See Figure 5).

GFP-tagged talin, paxillin, and vinculin have also been used productively for decades in the adhesion community. While one must be cautious tagging any protein, we use constructs that have been employed extensively in other labs and have been accepted as faithful reporters of adhesion formation or maturation. We also agree that overexpression is a valid concern. However, as described in the original manuscript, we transfected cells with a minimal amount of DNA (200 ng), or transduced cells with lentiviral vectors with a truncated CMV promoter that reduces expression 100-fold. In both cases, all cells were sorted for minimal expression (e.g., just greater than autofluorescence) with a flow cytometer just prior to imaging. Importantly, we observed quantitatively similar results for adhesion classification and maturation under talin knockdown and knockout conditions, for rescue experiments involving GFP- and mNeonGreen-tagged talin. Thus, our results are independent of the fusion protein, or whether or not endogenous talin is present.

Regarding the talin1 knockdown: is it know if talin2 is expressed in ChoK1 cells?

To evaluate the expression of different talin isoforms in ChoK1 cells, we performed a western blot for talin1 and talin2 in both WT and talin1 knockdown cells. Here, we find that talin2 is expressed at very low levels in WT and talin1 knockdown cells. While we cannot exclude some compensation between isoforms, the fact that we observe robust defects and proper rescue upon expression of talin1 variants suggests that interference by talin2, if any, is minor. Likewise, experiments in IMCD cells, where talin1 was rescued, but not talin2, also suggest that crosstalk between isoforms is negligible. These western blots are now provided in Figure 5—figure supplement 2 and discussed in the text.

Did the authors observe differences in the expression levels of talin1 wt-mNG or talin1 R8vvv-mNG?

As previously mentioned, all cells were evaluated and sorted with a flow cytometer just prior to imaging. Here, the intensity distributions of cells expressing wt-mNG talin1 and R8vvv-mNG talin1 were indistinguishable. Thus, we have no reason to believe that these variants are differentially expressed or degraded.

Why is the mNG-tagged talin1 running at a seemingly similar MW then the endogenous talin1 in Figure 5—figure supplement 1?

The molecular weight of talin1 and mNG are 270 and 26.6 kDa, respectively. As such, tagging talin1 with mNG only yields a 10% increase in molecular weight. Furthermore, in SDS-PAGE, electrophoretic migration scales logarithmically with the molecular weight of the protein. For these reasons, we anticipate that the tagged protein will have a very similar electrophoretic mobility. Nonetheless, we now include in Figure 5—figure supplement 1 the western blot results from WT cells overexpressing mNG-tagged talin 1 or mNG-talin R8vvv, which clearly show that a minor separation exists between endogenous talin 1 and mNG-tagged talin 1.

3) Their data demonstrate that the talin1 R8vvv mutation, which prevents talin-vinculin pre-complex formation, restricts NAs from maturing into focal adhesions. Is there a way to increase talin-vinculin pre-complex formation (e.g. by expressing an activated form of vinculin) to increase the number of maturing (G2) NAs?

We thank the reviewer for this suggestion. To increase the formation of a talin-vinculin pre-complex, we would need to identify a vinculin mutant with a higher affinity for R8. Unfortunately, we are not aware of such a mutant. There is a constitutively active vinculin mutant (T12) that promotes mature, over-sized adhesions, but it interacts with talin1 vinculin binding sites with its head domain (Cohen, D.M., Chen, H., Johnson, R.P., Choudhury, B., and Craig, S.W. (2005), J Biol Chem), which is not necessarily specific to R8. Thus, a potential experiment with the mutant will obviously induce more maturing NAs, but it would not test whether the pre-complex formation is R8-specific.

To support their hypothesis of the importance of pre-assembled talin-vinculin complexes it would be interesting to test the effect of talin1 variants impaired in actin binding to determine the importance of these independent actin linkages. What has a stronger effect?

We agree with the reviewer; it is a fascinating question whether talin R8 – vinculin – actin bridges could functionally replace the direct talin – actin bridges. However, we want to clarify, this is not our suggested model. As the cartoon in Figure 8 illustrates, we envision that both bridges have to work in parallel to support the formation of a maturing adhesion. Although we cannot exclude that in the very earliest stages talin R8 – vinculin – actin bridges might be sufficient to raise the traction above the levels seen measured in talin R8vvv expressing cells (i.e. the R8 – vinculin – actin bridges are more important than the talin – actin bridges), it is well established that the absence of the talin ABS causes a significant defect in adhesion formation and maturation that will likely make a G1 / G2 distinction very challenging. We point out to the reviewer that these experiments are tedious and given the reduced work capacity with the COVID pandemic we had to choose between the “necessary” and the “interesting”. We hope the reviewers and editor understand that we focused the continuation of the originally reviewed version of the study on bolstering the key novelty of our data: A talin – actin bridge alone is not sufficient to trigger mechanical adhesion maturation but the process requires additional early support from vinculin – actin bridges that bind to a still folded talin configuration.

4) The authors write that their analysis suggests, "that a very large number of adhesion protein aggregates, detectable through either talin, vinculin, or paxillin recruitment, never engage with the substrate." Is it not an overstatement to say that the failure to detect traction forces shows that these adhesions are not engaged with the substrate? Could they engage with the substrate but not link to actin?

We agree with the reviewer and have changed the wording appropriately. It now states that “a very large number of adhesion protein complexes, detectable through either talin, vinculin, or paxillin recruitment, do not transduce measurable traction forces (e.g., less than 20 Pa). Indeed, these complexes could result from an incomplete molecular clutch and be engaged with the substrate but not actin, or vice versa.”

Reviewer #2:Han et al. describe a pre-complex of talin and vinculin formed in the cytosol, in which the vinculin vd1 domain interacts with the talin R8 domain. The authors demonstrated this complex with recombinant protein fragments of talin and kindlin. An R8vvv mutation of talin mechanically stabilizes R8, reduces the formation of the pre-complex and the number of maturing NAs in cells. The authors used high resolution traction force microscopy and TIRF to conclude that the talin-vinculin pre-complex is essential to mature NAs and sustain traction forces applied to them characterized. While the data and the model potentially provide a functional role of R8 in NA maturation, several questions arise.1) To better understand the experimental setup, a figure explaining the method as well as an image of the flexible substrate with embedded beads measuring traction forces (including the processing steps) would be helpful. Additionally, the bead density and average distance between beads in these substrates should be reported. The mechanical properties of the substrates should be evaluated, as they might differ from batch to batch. Technically, I am wondering where the different background traction levels (see for example Figure 2, different traction force traces) or the traction measured outside a cell (see Figure 5A-C) arise from.

We now provide an explanatory figure that describes the traction force microscopy procedure (Figure 1—figure supplement 1A, E-H), as well as a detailed analysis of our bead density (~2.2 beads/µm^2^ or 0.42 ± 0.17 µm bead-to-bead spacing). Please see the new Figure 1—figure supplement 1B-D.

Every batch of substrates is evaluated prior to use, and substrates are only accepted if the stiffness falls within 10% of the nominal value.

We also discuss possible sources of the background traction. Conceptually, traction outside the cell may originate from non-transfected, invisible cells. However, this is rather unlikely, as cells were sorted with a flow cytometer prior to imaging. Alternatively, differences in background traction levels and traction forces outside of the cell are much more likely to result from imperfections in the image registration after removal of the cells: Per experiment ~15 cells were imaged for 10 minutes each. Once imaging was complete, all cells were removed by gently adding a dilute bleach solution that was preheated to 37 degrees Celsius, and the relaxed substrate was measured as rapidly as possible to avoid swelling of the substrate. Consequently, the entire experiment took ~2.5 hours, required careful fluid handling, and accurate positioning of the sample by the microscope. It is quite possible that certain parts of the substrate exhibit local deformations that are not captured by the registration procedure. Importantly, these spatial variations in traction force do not affect our local, temporal fluctuation analysis results.

2) Talin and vinculin are known to exhibit auto-inhibited conformations, in which an interaction between R8 and Vd1 may not be possible. Can the authors show in a more physiological setting (preferably full-length proteins) that talin and vinculin form a complex in solution?

We understand the reviewer’s concern that the R8-Vd1 interaction might not be possible in full-length talin and vinculin, especially in their autoinhibited state. However, it is not feasible to test the full length proteins in this experiment as there are 11 VBSs, and of these, the R3 VBSs are easy to activate (if we use Vd1 it can break open R3 as well). In an effort to support our argument, we now cite a recent study from the Ballestrem group. In this manuscript, they show that full-length talin does not associate with full-length vinculin unless the proteins are activated (Atherton et al., 2020). This, as well as thermodynamic principles, implies that a small population of talin or vinculin exists in an active conformation that is capable of forming a complex. In our manuscript, we speculate that such activation could potentially result from F-actin-driven force or vinculin’s PIP2 binding. While this possibility would be difficult to test experimentally, we have included this discussion in the text.

Additionally, in Figure 4C it seems that talin1 R8vvv can still bind vinculin, although not as proficient. Can the authors show the importance of vinculin binding to R8 for the maturation of NAs into FCs more directly by mutating only the vinculin binding site in R8?

While we agree that the R8vvv mutant can still bind vinculin, the equilibrium is shifted towards the unbound state (Figure 4C). Mutating the VBS alone in R8 is a good suggestion. Unfortunately, our attempts at making a mutant that prevents the VBS binding were not successful, i.e., all mutants tested did not abolish vinculin binding. The main challenge we faced in generating such a mutant was maintaining the structural integrity of the folded R8 domain, which is integral to ligand binding and must remain unperturbed. Indeed, these vinculin binding epitopes are buried in the folded form, and mutations appeared to easily ruin the fold. We attempted to alter the sizes of the hydrophobic epitope (swapping the F-A) but were not successful here either. We hope the reviewer understands these challenges.

3) The authors state that in maturing (G2) NAs vinculin and talin are recruited together. However, in Figure 2O it appears that in about 25% (and in Figure 7J ~50%) of cases, vinculin is recruited before talin. In Figure 7H it appears that vinculin is recruited to NAs 14 s earlier than talin. Could the authors elaborate how vinculin is recruited before talin is visible?

We appreciate the reviewer’s observation, and we revisited our data and looked closely at the trajectories where vinculin recruitment was identified earlier than talin. In these adhesions, we found that the talin signal was present but spatially diffuse and was thus not detected as a distinctive spot with our particle detection software. In contrast, although the vinculin signal was dim, it was spatially concentrated and detected by the software. Consequently, the detection of the early rise in talin was delayed relative to vinculin our analyses.

To evaluate if we could more accurately detect the onset of the diffuse talin signal, we relaxed the statistical criteria for particle detection from a P-value of.05 to.1 (e.g., 1 in 10 chance of it being observed due to chance). After reanalyzing the data, the number of events where vinculin arrived before talin was markedly decreased (see Figure 7N). In an effort to better communicate this observation, we updated Figure 7J and replaced Figure 7H with a new time series pair that represents a time-lag of 0 seconds, as now provide montages that follow adhesion assembly.

With regard to Figure 2O, we find that the larger pre-traction rise time for vinculin (than talin) is due to heterogeneity in traction background, which happens depending on multiple factors (e.g., image quality, bead density, and overall deformation, which can be all heterogeneous). In an effort to improve the quality of these data, we collected more samples from each condition and have analyzed the time of recruitment (updated in the new Figure 2O). It is worth noting that the negative portion of the 25% percentile is similar between talin and paxillin, and that the overall trend didn’t change with the new analysis.

4) How do the authors exclude the possibility that talin is quickly unfolded (as described in the legend of Figure 8) by forces from the cytoskeleton, which induces a fast recruitment of vinculin to talin in maturing NAs (which may be dependent on R8)? This process could be faster than the sampling rate and hence a pre-complex is not required to form in the cytosol.

We appreciate the alternative scenario that the reviewer has proposed. Indeed, we cannot formally exclude the possibility that vinculin recruitment is faster than our sampling rate. In part, this is due to the limitations of microscopy. Here, we had to image fast enough to properly detect and track adhesion complex formation, at low enough probe concentrations as not to perturb the specimen, with laser powers that did not noticeably introduce photobleaching or phototoxicity, all the while imaging long enough to watch the relatively slow process of adhesion maturation. Nonetheless, our observation is in agreement with previous work that used fluorescence fluctuation analysis to evaluate adhesion assembly, which is a method that provides a faster temporal sampling (see Bachir et al., 2014).

In contrast, testing the possibility by molecular perturbation might be informative if the effects of adhesion maturation are approximately equal. For example, such an experiment could include a double mutant R8vvv and ABS3 KO. Nonetheless, our data suggests this is not the case and gives overwhelming evidence that the key factor is R8 domain. Indeed, it is the only thing that can explain the differences observed in the IMCD talin null rescue experiments. Given that these experiments would require a significant amount of time and resources we chose not to pursue them (see also response 3-2 to Rev #1).

5) The authors show multiple traction force maps and images acquired for cells that show very different phenotypes although being the same cell line. In Figure 1A-F ChoK1 cells are shown that overexpress mGFP-tagged talin, vinculin or paxillin but have very different morphologies, sizes (d-f) and exhibit very different traction forces (a-c) where the scale is 0-300 Pa for talin, 0-1800 Pa for vinculin and 0-1000 Pa for paxillin. In Figure 2 it is similarly diverse where traction force scales range from 0-800 Pa (in h) to 0-4000 Pa (in a). What is the reason for this heterogeneity in morphologies and traction forces within the same cell line?

We understand the reviewer’s concern. However, we believe that these images reflect the natural heterogeneity of this cell line. Indeed, in an effort not to bias our results and to maximize the statistical power of our subsequent analyses, every cell encountered in the dish was imaged regardless of their morphology.

With regard to overall traction and cell area, we show these distributions in (Figure 1—figure supplement 2). It is worth noting that while there was slight variation in each protein-overexpression condition, no statistical difference in the spread area and overall traction was observed between the protein groups.

6) In maturing (G2) NAs, does the constantly increasing traction force arise from the accumulating more vinculin at the adhesion site (i.e. stronger connection to the actomyosin), or from the growth of the adhesion site (i.e. accumulating more integrins)? In Figure 2G, a continuous recruitment of talin is observed, which would argue for the increase in traction forces due to the recruitment of additional integrins. How would the authors explain the pre-formation of talin-vinculin complexes to be important for the recruitment of integrins to the adhesion site (in maturing NAs)?

We appreciate the reviewer’s suggestion of another potential mechanism for traction increase within G2 adhesions. We believe that both scenarios are plausible. As the reviewer noticed, the time series of talin in WT G2 adhesions steadily increase during traction development (shown in both Figures 2G and 6H), which can be proportional to the number of clustered integrins. If we compare the (original) Figure 7B,D to 7F,H, where talin and vinculin are imaged simultaneously, we observe the following:

1). For wild-type talin – talin’s intensity remains relatively constant in G2 adhesions (t=200-600 sec in Figure 7D), whereas vinculin’s signal increases significantly (t=200-600 sec in Figure 7H).

2). For the R8vvv mutant of talin – talin’s intensity shows a large increase in G2 adhesions (t=82-280 sec in Figure 7B), whereas vinculin’s signal increases marginally (t=82-280 sec in Figure 7F).

In Figure 6B, the traction of R8vvv G2 adhesion shows a plateau after 400 sec while talin signal was exponentially increasing, implying that inability to incorporate further vinculin could result in failure of further traction development. This gives more weight to the evidence that the precomplex formation has a major effect on vinculin recruitment.

7) For testing the effect of the R8vvv talin mutant in cells, the authors used shRNA to knock-down talin1 levels before rescuing cells with either WT or R8vvv talin1. How is the talin expression level of WT talin1 rescued cells compared to R8vvv rescued cells?

We sorted cells with a flow cytometer prior to imaging, and both the R8vvv and wild-type cells had similar expression profiles before and after sorting (see also response to Rev #1 comment 2).

We also evaluated talin1/2 double knockout IMCD cells (from Dr. Roy Zent) to investigate R8vvv’s effect in a cleaner manner (e.g., in the absence of residual talin1 expression, as well as talin2 expression). Indeed, the adhesion phenotype in these cells was in agreement with the ChoK1 cells. In particular, the R8vvv mutant showed significantly less adhesion maturation (Figure 5—figure supplement 4).

Further, an assessment of the morphology (whole cells and possible adhesion sites) of talin1 KD cells would be beneficial to appreciate the difference between R8vvv and WT talin 1. This would exclude the possibility that apparent differences between R8vvv and WT talin1 re-expressing cells arise from potentially different talin1 expression levels, a potential expression of talin2 in response to reduced talin1 levels or residual expression of WT talin1 in KD cells.

We reproduced our key findings from ChoK1 cells in talin1/2 double KO IMCD cells. Consequently, this should eliminate the concern that the differences observed between R8vvv and WT talin1 rescue cells are due to variable talin1 or talin2 backgrounds. As replied to reviewer #1, the expression levels of R8vvv and WT talin1 were controlled via judicious transfection/transduction conditions and by sorting cells with a flow cytometer prior to imaging. Likewise, as we previously stated, we believe that the morphology of the cells reflects the natural heterogeneity of the cell line, and that we intentionally chose to image all cells regardless of their morphology as to not bias our analysis.

8) The authors state that the talin-vinculin pre-complex formation is reduced in talin1-R8vvv expressing cells. However, talin1-R8vvv expressing cells show adhesion sites larger than NAs (Figure 5A). How do larger adhesion sites form in cells expressing talin1 R8vvv, if the vinculin binding to R8 is essential for NAs maturation? Representative image of labeled vinculin and talin within one cell (as analysed for Figure 7) would be beneficial to understand how the vinculin recruitment is affected by the R8vvv mutation.

This is a valid observation. To investigate the possibility that the large adhesions in talin1-R8vvvexpressing cells are driven by residual endogenous talin1, we investigated the adhesion pattern in talin1/2 double knock-out IMCD cells with talin-R8vvv expression. Consistent with the ChoK1 cells, these cells also show fewer maturing NAs than control cells, but also some large adhesions. There are two non-mutually exclusive explanations for this: (1) talin-R8vvv still can bind vinculin or (2) that vinculin-talin precomplex formation is not the only path for adhesion maturation. Indeed, in support of the former, we show evidence in Figure 4C that binding is reduced, but not abolished, for talin-R8vvv and vinculin. Nonetheless, given the complexity of cell adhesions, it seems reasonable to assume that secondary mechanisms for adhesion maturation also exist.

In order to better communicate how vinculin recruitment is affected by the R8vvv mutation, we now include representative images of vinculin and talin for both R8vvv and WT talin1 expressing cells (new Figure 7). In agreement with reduced talin-vinculin binding, R8vvv cells do have G2 adhesions, albeit less of them. Possibly, there is a compensation effect at play where less abundant G2 adhesions accumulate more talin. We also show that vinculin recruitment in those G2 adhesions is indistinguishable compared to one in G1 adhesions in R8vvv expressing cells.

9) In their very clear conclusive Figure 8 the authors depict different sites for paxillin in non-maturing vs. maturing NAs. Do the authors have experimental evidence for this?

The sites depicted are based on findings in the literature, not from our experimental findings. We now added question marks on every paxillin’s appearance in the figure and added our explanation in the figure legend with appropriate citations.

Further, the legend describes hypotheses, which can explain experimental data but experimental evidence from the reported data is limited. For example, that talin stretching is limited to a shorter level in non-maturing NAs and talin stretches faster in maturing NAs has, to my point of view, not been shown in the presented data.

We agree with this critique and have toned down our interpretation.

Reviewer #3:Han et al. describe in great detail the molecular decision process that leads from transient nascent adhesions (NAs) forming constantly at the plasma membrane to focal adhesions (FAs) stably connecting the ECM with the actin cytoskeleton. The assembly kinetics of three GFP-tagged NA components (talin, vinculin and paxilin) is tracked down by TIRF microscopy and their behavior is correlated with local traction force determined by imaging the displacement of fluorescent beads in the hydrogel support, on which the cells grow. The authors apply machine learning algorithms to identify two classes of events both linked to force-assisted growth but with opposite fates. Using a new talin mutant, it is shown that binding of vinculin at a specific talin domain (R8) is required to establish the feed forward control for further vinculin recruitment and the steady build-up of local traction. In contrast, sequential binding of talin, vinculin and paxilin leads to maturation abort. I consider the study to be compelling, the technology pioneering and the results significant. Given that the topic is quite complex, the manuscript is of remarkable clarity. A direct comparison with R3 talin-mutants would add further evidence, however, such experiments are probably not within the scope of a revision. My listed points represent mainly comprehension questions that could encourage improvements of the text. Overall, I am in full support of publication.1) Is there statistics on how many G2-class NAs actually mature into FAs?

We thank the reviewer for the support. Indeed, a direct comparison with the talin1 R3 IVVI mutant would be of significant interest. As the reviewer noted, this requires a substantial investment in time and resources. Especially with our reduced lab capacity, we believe that this may take ~1 year to achieve. As such, we believe it is beyond the scope of this publication.

2) "Event-based time series analysis" measures the appearance of fluorescently tagged proteins. What is the sensitivity of the method? How many mGFPs have to accumulate to detect a spot?

We appreciate the reviewer’s insightful questions. To evaluate the number of GFPs necessary for detection, we performed protein number and brightness quantification (Digman et al., 2008 Biophys J) on a single movie (i.e., paxillin-mGFP). Using such an approach, we find that the minimum number of mGFPs necessary to detect NAs is ~2 (see Author response image 1).

**Author response image 1. sa2fig1:** Histogram of mGFP molecules in one paxillin-mGFP time lapse images. Mean value was 1.7 with 1.08 standard deviation. Up to 16 mGFP molecules were able to be resolved.

3) At the stage of Figure 2, is the existence of an unstrained talin-vinculin pre-complex really a stringent conclusion? The time resolution of imaging is one second. During this time bin, proteins with a Kd in the μM range may undergo several binding and unbinding cycles. Likewise, the forces may exhibit fast fluctuations especially during the onset when only a small number of force transmitting proteins have been accumulated (see Figure 3B). Thus, there is plenty of "temporal" space for different scenarios of "what is first" that is technically inaccessible. Please comment.

We agree with the reviewer’s thoughts. During our imaging experiments, we carefully balanced our imaging duration, image acquisition rate, and illumination conditions, such that we could simultaneously resolve both fast (molecular recruitment) and slow (e.g., maturation) adhesion processes. At our purposefully minimal expression levels, a frame rate of 1 Hz was the fastest we could achieve without introducing photodamage. Consequently, we would not be able to resolve phenomena that occur on sub-second timescales. Nonetheless, our results are consistent with previous work that used very fast fluctuation-based imaging methods (see Bachir et al., 2014).

In order to communicate this limitation to the readers, we have reworded the text. For example, we use “concurrent localization” instead of “pre-complex formation” in the paragraphs up to Figure 3. Only after Figure 4, where we show a complex between talin R7R8 fragment and vinculin head, do we begin using the term “pre-complex”.

4) Figure 3B, the average force growth in the first 10 seconds may simply be zero due to the sensitivity limit of TFM. Or does this represent a finding?

We appreciate the reviewer’s insightful point. We have analyzed the background fluctuation of the traction and found that the traction growth in the first 10 seconds is not significantly different from the traction background fluctuation. We have updated this point.

5) Are G1-NAs programmed to terminate, because talin has too much time for erroneous interactions? Explicitly, do traction-dependent vinculin binding site (VBS), other than R8, represent "wrong sites" in the sense that cooperative vinculin assembly is even prevented?

We are grateful for this interesting interpretation regarding G1 NAs. We haven’t explicitly tested this hypothesis because of the complexity involved in the experimental design. Nonetheless, we now add a paragraph that outlines potential scenarios for G1 NAs in the Discussion.

6) At which stage of NA maturation then must force be sensed?

We suggest that the force should be sensed at the earliest stage of the NA assembly. We have made this point clear in the text describing the results in Figure 2O.

7) Is mutant R8vvv-talin totally impaired in vinculin binding (Figure 4D) or is there residual probability for exposing the VBS (Figure 4C)? The biochemical result is contradictory.

To resolve this contradiction, we repeated binding experiments with multiple variants of vinculin, including:

1) Vd1

2) A vinculin mutant (A50I) deficient for talin binding

3) A constitutively active vinculin mutant (T12)

4) Wild-type full-length vinculin.

Unfortunately, these experiments suffered from a high level of variation between repeats, and we were not able to identify the source of this variability. As such, we have chosen not to include them. Nonetheless, as we show in Figure 4C, the Vd1 domain of vinculin can bind the R8vvv mutant of talin, albeit to a lesser extent. For clarity, we have removed Figure 4D.

8) The difference between vinculin epitopes in domain R3 and R8: As a naïve reader one would assume that a VBS of the relaxed state locates at the surface of the protein. I consider it conceptually difficult to acknowledge that the VBS in domain R8 is active in an unstrained state but nevertheless requires unlocking of the threonine belt and partial unfolding of the α-helical bundle; maybe this warrants a sentence or two.

Both R3 and R8 have similar mechanical responses. For example, both domains are folded, but destabilized by their threonine belts. Consequently, although the VBS is not always accessible, the presence of the threonine belt shifts the equilibrium towards an accessible state.

From the literature, we know that full-length vinculin cannot bind to full-length talin unless one of the proteins is in a relaxed state (see Atherton et al.). As such, we hypothesize that a spontaneous conformational change (perhaps to an intermediate or semi-relaxed form) could release talin autoinhibition, expose the R8 VBS, and permit vinculin binding.

With regard to R3 in particular, the same study showed that talin lacking R2R3 is able to interact with both inactive and active forms of vinculin. Nonetheless, why vinculin cannot bind to the R3 domain in the absence of force is an open question in the field. One possibility is that RIAM, which binds to the R2R3 domain with a high affinity, stabilizes R2R3 in a conformation where the VBS is in a less accessible state. From our data, we have no reason to believe that such a masking interaction occurs on R8. In an effort to communicate these nuances to the reader, we added a paragraph in the Discussion on the role of R8 in VBS accessibility.

9) Cells expressing the R8vvv mutant are clearly impaired in NA maturations as documented by the cellular analysis (Figure 5). However, there are mature FAs. This may either relate to residual wildtype talin that could not be suppressed by shRNA (as the WB shows, Figure 5—figure supplement 1), residual binding of the mutant R8vvv-talin or even an undefined pathway into the cooperative vinculin assembly. These possibilities are not mentioned.

We appreciate the reviewer’s observation and potential interpretations, which were also raised by Reviewer 2. To investigate the possibility that the large adhesions in talin1-R8vvv-expressing cells result from an incomplete knock down of talin1, we performed rescue experiments in talin1/2 double knock-out IMCD cells. Importantly, rescue experiments in the IMCD cells phenocopied the experiments performed in ChoK1 cells. Specifically, cells rescued with the R8vvv mutant of talin1 have fewer maturing adhesions than cells rescued with WT talin1, but continue to display some large FAs (e.g., Figure 5J).

Another possibility is that these large FAs could be promoted by talin2. However, because the IMCD cells lacked talin2 expression but still created large FAs, this can be ruled out.

Thus, we think that residual binding on the mutant R8vvv-talin to vinculin is likely to explain these large FAs. As we show in Figure 4C, binding is reduced, but not abrogated, for talinR8vvv and vinculin. Nonetheless, given the complexity of cell adhesions, it seems reasonable to assume that secondary mechanisms for adhesion maturation may also exist. We now discuss these possibilities in the Discussion.

10) The traction growth rates in Figure 6K reach much more into the negative range than those in Figure 3B and 3C. Do negative values signify initial forces that disappear? In addition, there is quite a large difference in the scale of y-axis in Figure 6K (100-200 Pa/sec) as compared to Figure 3B and C (1-2 Pa/sec).

We have verified the value of the data and parameters that were used to quantify those data.

First, there was a trivial error in the y-axis labeling: The unit in Figure 6K was meant to be “Pa/min”, “Pa/sec” as indicated. The labels in Figure 6K and 3B,C are now consistent.

The differences in force growth rate between Figure 6K and Figure 3B-C can be easily explained. Specifically, in Figure 6K, we evaluated all adhesions (i.e., those in both G1 and G2). In contrast, in Figure 3B-C, we evaluated only “force-transmitting” adhesions. To avoid this confusion, we now present only the “force-transmitting” NAs in Figure 6k as well.

Of note, we chose to analyze the force-transmitting adhesions since this served as the best mechanism to identify de novo NA nucleation events. Furthermore, because we could align NA assembly to force onset, this allowed us to perform the time-delay analysis as shown in Figure 2M.

11) The body text referring to Figure 6 at the end of the paragraph says "These results suggest that the vinculin-talin pre-complex is essential for the development of force across NAs…". In my opinion this statement still remains an assumption, since I could not find direct evidence in Figure 6. The behavior of the mutant R8vvv-talin, if G2 characteristics are fulfilled, appear to be indistinguishable from the mGFP-tagged version described in Figure 2. The fact that there is a problem with force growth can only be explained in the light of Figure 7. Re-ordering these arguments may be considered.

We agree that the problem with force growth in G2 adhesions of R8vvv mutant cells can only be explained with evidence from Figure 7. As we observe the different force development in the G2 adhesions, we now ended the paragraph with a more direct summary of the observation.

12) Again, the two expression systems lead to a different range of values: The absolute vinculin assembly rates in Figure 7I as well as the rate increase between G2 vs G1 are much smaller than in Figure 3A. Is this due to different protein expression levels? Why were different integration times used (30 seconds in Figure 7I in contrast to 10 seconds in Figure 3A)?

In Figure 7, we used the amplitude of a Gaussian fit to approximate the experimental point spread function, which provided a noise-suppressed “signal” and allowed us to compare the two fluorescence images.

In Figures 2 and 3, where we compare fluorescence vs traction, we used the absolute intensity since it was enough to compare the relative difference.

To avoid the confusion, we now also report the amplitude of the Gaussian fits in Figures 2 and 3. We also apply the same “first-rate constant method” for Figure 7 as in Figure 3.

13) Is actin always engaged first and then the integrin tails? In other words, is there support for a "search mode" of a small number of competent actin-talin-vinculin-heads for integrin tails linked to the ECM (Figure 8, G2, first panel)?

No, we have no evidence that actin is always engaged first. Furthermore, we only have indirect evidence in the literature of actin engaging with the talin-vinculin complex. In an effort to communicate this graphically, we have added question marks on the assembly of the “actin” part in the cartoon Figure 8. We hope this clearly indicates that this part of the illustration is not supported by our data.

[Editors’ note: what follows is the authors’ response to the second round of review.]

Essential revisions:1) The conclusion that the concurrent recruitment of talin and vinculin to maturing NAs corresponds with the existence of a cytoplasmic precomplex of the two proteins. Although it is likely that such precomplexes form prior to NA recruitment, it has not been demonstrated in the current paper. Hence, it is necessary to tone this claim down in the title, Abstract, Result sections and Discussion of the manuscript.

We agree that our data does not provide direct evidence of a cytoplasmic precomplex, and we have sought to tone down this claim throughout the manuscript. In particular, we have replaced the word “precomplex” with “concurrent recruitment” or “force-free complex formation” in Abstract, Results, and Discussion. To generate a succinct summary of the main result in the title, we use the phrase “pre-complexation of talin and vinculin without tension”. This wording avoids the unproven notion that talin and vinculin come into the adhesion as precomplexes but emphasizes that the two molecules are recruited and interact, at least in adhesion sites, before tension develops. We hope that this is an appropriate tone-down.

2) Use same assembly rates (Figure 3A, Figure 6J, and Figure 7M) in y-axis. It is not clear what is meant by 1/min in Figures 3A/7M, and what is the rational for the different time frames analysed (Figure 3A: 2 min; Figure 6: 1 min; Figure 7: 0.5 min).

We thank the reviewer for catching the differences. The unit “1/min” was meant to be a first-order rate constant. However, to be consistent with our calculation of force growth rate, we decided to use the slope of the intensities. Accordingly, we changed the unit in Figures 3A and 7M to be “a.u./min” with correctly converted values. Importantly, these values are now comparable with Figure 6J.

For our analysis, we used a 10 second integration time for Figure 3A and a 20 second integration time for Figures 6J and 7M. In Figure 3A, we used a shorter integration time because it nicely demonstrated that vinculin had a substantially different association rate for non-maturing and maturing nascent adhesions. However, these results would not change markedly when analyzed with a 20 second integration time. In Figure 6J and 7M, we found that the longer integration time produced less variance in the calculated slopes. To better communicate these details, we have fixed the time information in the figure legends of figures 6J and 7M accordingly.

3) Although the authors state that the R8vvv mutation does not affect the talin recruitment rate, Figure 6J shows that in both G1 and G2 NAs talin R8vvv is recruited at higher rates than WT talin1. Test the rates for statistical significance (for G1 and G2 R8vvv vs. WT). In case talin1 R8vvv is indeed recruited faster to adhesion sites, the authors should try finding an explanation.

We thank the reviewer for noticing the potential difference in talin recruitment between R8vvv and WT talin. Indeed, there were statistical differences between the two conditions, in every combination of G1 and G2, which we now show in Figure 6J. Presumably the higher rate of R8vvv recruitment is consistent with the overall increased rate of NA nucleation. But we have no molecular explanation for this phenomenon. We tend to avoid grandiose speculations and thus would like to leave the presentation as is with the facts statistically pinned down. However, as can be seen in the adhesion trajectories, G1 adhesions under the R8vvv condition assemble fast in the first minute but also disassemble fast. Under these conditions, vinculin is not recruited significantly. To explain this to the readers, we added this explanation to the manuscript and now include an example (Figure 6—figure supplement 1).

4) The authors state that traction increase for talin1 R8vvv is reduced compared to WT talin1. The statistical test in Figure 6K to corroborate this statement is missing.

We added bars in Figure 6K that show statistical differences with asterisks.

5) Subsection “Simultaneous talin-vinculin imaging confirms vinculin’s recruitment after talin for non-maturing NAs and concurrent recruitment for maturing NAs”: should probably reference Figure 7N (instead of Figure 7J) and Figure 7O (instead of Figure 7K).

We corrected the figure panel and references accordingly. Thank you.

6) "Fluorescence fluctuation analysis" may be misleading as it usually refers to FCS, PCH, i.e., methods that involve a statistical correlation analysis. Here the authors directly analyze the time traces by thresholds and slopes.

We believe that this comment is valid, and have changed the wording to “event-detection time-series analysis”.

7) The density 2.2 beads/µm2 cannot be reproduced from average spacing 0.42 µm via the relation: density = 1/sqrt(spacing).

We used a formula, i.e., density = detected beads/area, to quantify the density, which was different from the relation given. However, we noticed that the given formula is more widely accepted. Thus, we changed the density accordingly and reflected in the manuscript and corresponding Figure 1—figure supplement 1.

8) "Both peptides bound…" This sentence sounds strange and may be changed to "Both peptides showed comparable affinity towards wildtype R7R8 and the R8R7vvv mutant, confirming.…"

We have adopted throughout the manuscript. Thank you.

9) "The threonine belt in talin R8 is responsible…" The wording is not clear since only the binding epitope itself can be responsible for binding and the threonine belt is, rather indirectly, regulating accessibility and hence the on-rate. Do these data suggest that RIAM and DLC1 use binding epitopes spatially distinct from the VBS within the R8 bundle? Or is RIAM and vinculin binding indeed competitive?

Thank you for pointing this out. We agree that the sentence was not sufficiently clear. Indeed, given that both RIAM and vinculin are capable of binding the VVV mutant form, the binding epitope remains unchanged. Instead, the VVV mutant has an altered bundle stability. In order to better communicate this, we have changed the wording of the corresponding sentence to: “Altogether, these biochemical characterizations suggested that the threonine belt in talin R8 is responsible for the increased accessibility of the VBS in R8 enhancing vinculin binding without force.” We hope this change clarifies the meaning.